



# Inferring the vertical distribution of CO and CO₂ from TCCON total column values using the TARDISS algorithm

Harrison A. Parker[1], Joshua L. Laughner[2], Geoffrey C. Toon[2], Debra Wunch[3], Coleen M. Roehl[1], Laura T. Iraci[4], James R. Podolske[4], Kathryn McKain[5,6], Bianca Baier[5,7], Paul O. Wennberg[1,8]

[1] Division of Geological and Planetary Sciences, California Institute of Technology, Pasadena, CA, USA

[2] Jet Propulsion Laboratory, California Institute of Technology, Pasadena, CA, USA

[3] Department of Physics, University of Toronto, Toronto, ON, Canada

[4] Atmospheric Science Branch, NASA Ames Research Center, Moffett Field, CA, USA

[5] Cooperative Institute for Research in Environmental Sciences (CIRES), University of Colorado, Boulder, CO, USA

[6] Earth System Research Laboratory, National Oceanic and Atmospheric Administration, Boulder, CO, USA

[7] Global Monitoring Laboratory, National Oceanic and Atmospheric Administration, Boulder, CO, USA

[8] Division of Engineering and Applied Science, California Institute of Technology, Pasadena, CA, USA

*Correspondence to*: Paul O. Wennberg (wennberg@caltech.edu) and Joshua L. Laughner (josh.laughner@jpl.nasa.gov).



**Abstract**

We describe an approach for determining limited information about the vertical distribution of carbon monoxide (CO) and carbon dioxide ($CO_2$) from total column observations from ground-based TCCON observations. For long-lived trace gases, such as CO and $CO_2$, it has been difficult to retrieve information about their vertical distribution from spectral line shapes in the shortwave infrared (SWIR)
spectra because of the large doppler widths at 6000 $cm^{-1}$, and errors in the spectroscopy and in the atmospheric temperature profile which mask the effects of variations in their mixing ratio with altitude in the troposphere. For $CO_2$ the challenge is especially difficult given that these variations are typically 2% or less. Nevertheless, if sufficient accuracy can be obtained, such information would be highly valuable for evaluation of retrievals from satellites and more generally for improving the estimate of
surface sources and sinks of these trace gases. We present here the Temporal Atmospheric Retrieval Determining Information from Secondary Scaling (TARDISS) retrieval algorithm. TARDISS uses several simultaneously obtained total column observations of the same gas from different absorption bands with distinctly different vertical averaging kernels. Since TARDISS avoids spectral re-fitting by ingesting retrieved column abundances, it is very fast and processes years of data in minutes. The
different total column retrievals are combined using a Bayesian approach where the weights and temporal covariance applied to the different retrievals include additional constraints on the diurnal variation in the vertical distribution for these gases. We assume that only the near surface is influenced by local sources and sinks, while variations in the distribution in the middle and upper troposphere result primarily from advection that can be independently constrained using reanalysis data about the
variation in mid-tropospheric potential temperature. Using measurements from five North American TCCON sites, we find that the retrieved lower partial column (between the surface and ~800 hPa) of the CO and $CO_2$ dry mole fractions (DMF) have slopes of 1.001±0.002 and 1.007±0.002 with respect to lower column DMF from integrated in situ data measured by aircraft and AirCore. The average error for our CO retrieval is 0.857 ppb (~1%) while the average error for our $CO_2$ retrieval is 3.55 ppm (~0.8%).
We calculate degrees of freedom from signal of 0.218 per measurement for lower partial column CO on average and of 0.353 per measurement for lower partial column $CO_2$ on average. Compared with classical line-shape-derived vertical profile retrievals, our algorithm reduces the influence of forward model errors such as imprecision in spectroscopy (line shapes and intensities) and in the instrument line shape. We anticipate that this approach will find broad application for use in carbon cycle science.

**1 Introduction**

Remote sensing measurements of atmospheric gases are made around the world in an attempt to better understand the sources, sinks, and fluxes at the local, regional, and global scales (Connor et al., 2008, p.2; Deeter, 2004; Kerzenmacher et al., 2012; Wunch et al., 2011). Compared with in situ


measurements, these retrievals are less influenced by nearby point sources or sinks and rapidly changing
meteorological conditions (Keppel-Aleks et al., 2012) which is both a strength and a weakness for use
in carbon cycle science investigations. Additionally, because the column represents the integral of a gas
from the surface to the top of the atmosphere, flux estimates from column amounts are less sensitive to
errors in the assumed vertical transport than those using surface measurements (Keppel-Aleks et al.,
2011, 2012).

Contrastingly, column measurements have their own drawbacks for estimating surface fluxes.
Total column concentrations are much less sensitive to the local emissions and flux estimation can be
influenced by variation in the mixing ratio at higher altitudes. Signals of $CO_2$ and CO from the surface
are muted in the total column due to the dilution of signals from the surface being integrated across an
entire column. For $CO_2$, the total columns are strongly influenced by mesoscale flux patterns in the
troposphere above the boundary layer making it even more difficult to discern the influences of surface
fluxes (Keppel-Aleks et al., 2011, 2012). For CO, its several-week lifetime in the free troposphere
results in regional transport influences that can dampen the surface signals in the total column values
(Deeter, 2004; Zhou et al., 2019). These issues can limit the effectiveness of total column measurements
in surface flux analysis – particularly for local sources.

Profile retrievals can, in principle, ameliorate these issues and provide clearer information on
surface processes.  Theoretical analysis shows that two to three vertical degrees of freedom (DoF) can
be achieved in $CO_2$ retrievals from near-IR (NIR) and mid-IR (MIR) spectra from high-resolution
Fourier transform spectrometers (Connor et al., 2016; Kuai et al., 2012; Roche et al., 2021; Shan et al.,
2021). In practice, Connor et al. (2016) and Roche et al. (2021) showed that the precision of retrieved
$CO_2$ profiles using spectral windows in the NIR was much lower than the theoretical estimate due to
errors in the a priori temperature profile and in the forward model. Likewise, Shan et al. (2021) retrieve
$CO_2$ profiles using spectral windows in the MIR. They use an a posteriori optimization method to
improve the tropospheric $CO_2$ signal and they report errors near 2%. Although both of these methods
retrieve profiles with sufficient degrees of freedom to observe some signals of the variation in the
vertical distribution, they report errors which limit their use for carbon cycle studies.

        Several operational CO profile retrievals exist, but these products still face the issues of column
dilution or larger sensitivity to the free troposphere compared to the surface. For example, spectra
recorded from the MOPITT instrument aboard the Terra satellite in both MIR and NIR have been used
to provide limited information on the vertical distribution of CO (between one and two degrees of
freedom in the troposphere)  (Deeter, 2004; Turquety et al., 2008). The Network for the Detection of
Atmospheric Composition Change (NDACC) retrieves profiles of CO in the atmosphere (Buchholz et
al., 2017) with ~2 degrees of freedom for the signal providing information of a lower (surface-8km)
layer sensitive to the boundary layer and an upper (8-20km) layer with ~1-3% uncertainty in the total
column (Zhou et al., 2018, 2019). These measurements require higher spectral resolution and therefore
a longer measurement time, resulting in fewer observations per day. This limits their ability to capture





diurnal changes and makes the measurements more susceptible to cloud interruption. These measurements also require accurate knowledge of the spectral line widths, their temperature dependence, the instrument line shape (ILS), and the solar spectrum. These limitations motivate our work to develop a new product with better sensitivity to surface processes and higher temporal
resolution.

Profile retrievals that fit measured spectra and exploit the profile information given by pressure broadening of spectral lines require high resolution data to obtain information about different levels of the atmosphere (Sepúlveda et al., 2014). In the approach described here, we do not retrieve profile information directly from the spectra. Instead, we utilize information from the vertical, temporal, and a
priori vertical profile domains to infer partial column dry mole fraction values. We fit partial column scalar values to match TCCON total column dry mole fraction measurements that are 1) quality controlled and 2) individually tied to the WMO trace gas standard scale which mitigates a number of errors in the forward spectroscopic model. We use multiple total column measurements from spectral windows with different line intensities, and hence different shapes of the column averaging kernel. We
extract the vertical information from the differences in total column values between the different windows by fitting over an entire day of measurement in order to make use of the information from the temporal dimension. We optimize the separation between the near surface and the rest of the atmosphere using additional a priori information about the expected temporal covariance in the different partial columns based on known atmospheric behavior. This method allows us to use the algorithm to
extract information focused on the lower atmosphere where the concentrations are most sensitive to surface exchange.

The accuracy of this new method for retrieving partial column values is evaluated using comparisons with in situ vertical profiles. Section 2 describes the theory and parameters chosen for our retrieval, and the data used for the retrieval, validation, and comparison. Sections 3.1 to 3.2 present our
validation data, a sensitivity study of the retrieval parameters, and an error and information content analysis. Finally, Sect. 3.3 and 3.4 give examples of data using this approach and provide evidence for the utility of this approach in flux estimation.

## 2 Methods

### 2.1 Total Carbon Column Observing Network

The Total Carbon Column Observing Network (TCCON) is described by Wunch et al. (2011), although we will give a brief overview here to include aspects of the retrieval algorithm and observation scheme that have evolved since 2011. TCCON is a network of sites that use ground-based Fourier transform spectrometers with InGaAs and Si detectors to gather spectra for the 3900 to 15500 cm$^{-1}$



spectral region. Importantly for our work here on CO, some sites are now equipped with an InSb
detector that simultaneously allows spectral measurement down to 2000 cm$^{-1}$ at the expense of
simultaneous observations using the Si detector. $CO_2$ and CO are retrieved simultaneously over several
spectral windows (independent spectral bands). These windows are chosen to provide high sensitivity to
the gas of interest while limiting interference from other atmospheric absorbers.

Column abundances of atmospheric species are computed from the measured spectra using a
nonlinear least-squares fitting algorithm, GFIT, which minimizes the residuals between a measured
spectrum and one calculated by uniformly scaling a priori vertical profiles for the fitted atmospheric
species, yielding the optimal VMR (volume mixing ratio) scaling factors (VSF) of the fitted gases. The
prior profiles scaled by the VSF are integrated to calculate the total column abundance of a species. The
retrieved scaled column abundances are converted to column dry mole fraction (DMF) by dividing by
the column of $O_2$, retrieved from a different spectral window of the same spectrum. These retrievals are
then quality-controlled and scaled to minimize both any airmass dependence and the difference with in
situ profiles. These outputs from standard TCCON processing are used as input for TARDISS.

For each window and for each spectrum fit by GFIT, an associated column averaging kernel is
computed that describes the sensitivity of the VSF to changes in species abundance at each altitude. A
perfect column averaging kernel would have values of one for all altitudes. More commonly, the kernels
will be greater than 1 at lower/higher altitudes and less than 1 at higher/lower altitudes. Values higher
(lower) than 1 mean that the retrieval is more (less) sensitive to changes at that altitude. These
sensitivities also vary with solar zenith angle (SZA) as the spectral absorption deepen. The vertical
sensitivity of each window is a result of its spectral properties.  Optically thin spectral regions
(windows) tend to be more sensitive to the upper troposphere and the stratosphere while optically thick
windows tend to be more sensitive to the lower troposphere. Since information about the stratosphere
comes from near the line center as a result of collisional broadening, if the absorption at the line center
is saturated (nearly zero transmission), the spectrum will contain little information about the
stratosphere and hence the kernel will be small there. The differences in column averaging kernel
shapes are the main source of information used in our algorithm.





**Figure 1.** Vertical sensitivities of the total column retrievals from GFIT and used in our algorithm for both $CO_2$ (left and middle column) and CO (right column) plotted against normalized pressure and color coded by the solar zenith angle (SZA). We also use a window centered at 2111 cm$^{-1}$ for CO which has vertical sensitivities that are nearly identical to the 2160 cm$^{-1}$ window. A column averaging kernel greater than 1 means that the total column is more sensitive to molecules at this pressure level than the average sensitivity. For example, if we move some of the $CO_2$ from 200 hPa to the surface in our a priori profile, the retrieved scale factor (VSF) will decrease for the 6073 cm$^{-1}$ window and increase for the 4852 cm$^{-1}$ window while the true total column remains unchanged. The 6220 and 6339 cm$^{-1}$ $CO_2$ windows have near-identical kernels due to the $CO_2$ bands being almost identical in their line strengths, separations, widths, and temperature dependences.

Since the terminology for the fitting done in a standard TCCON retrieval is similar to that used in the partial column retrieval discussed in this work, we will refer to the standard TCCON retrievals as



being the phase 1 (P1) fit and the partial column retrievals as the TARDISS fit. We also use the terms retrieval and fit interchangeably to refer to the P1 or TARDISS methodology.

### 2.1.1 Sites Used in this Work

In this study, we use data from five TCCON sites located across the United States from as early
as 2004 to as recent as 2021. These are located at Park Falls, Wisconsin; NASA Armstrong, Edwards Air Force Base, California; Lamont, Oklahoma (the DOE Southern Great Plains ARM site), the California Institute of Technology (Caltech), in Pasadena, California, and East Trout Lake, Saskatchewan, Canada. Table 1 presents a summary of the sites used in this work.

Park Falls, WI hosts the first operational TCCON site (July 2004-present). The site is in a rural,
heavily forested area and generally far from anthropogenic influence. The FTS does not have an InSb detector so we are able to only retrieve partial column values for $CO_2$. We focus on data obtained since 2012, when the alignment of the instrument has been more consistent. The increased variance of the TARDISS retrieval for data before 2012 likely reflects the inconsistent alignment of the FTS.

We use similar data from the TCCON site located at NASA's Armstrong Flight Research Center
(formerly the Dryden Flight Research Center) in California which has been operational since July 2013. We report $CO_2$ partial column values for the 2013 to 2021 time period. The Armstrong site is on the northwest edge of Rogers Dry Lake within the Edwards Air Force Base in the Mojave Desert.

The Lamont, OK TCCON site is surrounded by farmland. It has been operational since July 2008, and an InSb detector was installed in October 2016. We report $CO_2$ partial column values from
2008 to 2021 and CO partial column values from 2017 to 2021.

The TCCON site on the Caltech campus in Pasadena, CA has been operational since July 2012 with an InSb detector measuring since October 2016. We report $CO_2$ partial column values from 2012 to 2021 and CO partial column values from 2017 to 2021.

The East Trout Lake, Sask., CA TCCON site is located in a remote, heavily forested area in the
middle of the Saskatchewan Province. The instrument uses an InSb detector so allowing us to retrieve partial column CO values. It has been operational since October 2016 and we report partial column values for CO and $CO_2$ from 2017 to 2021.






| Site | Location | Dates of Measurements Used | Data DOI |
|---|---|---|---|
| Park Falls, WI | 45.945N, 90.273W | $CO_2$: 2012 - 2021 | 10.14291/tccon.ggg2020.parkfalls01.R0 |
| NASA Armstrong, Edwards Air Force Base, CA | 34.958N, 117.882W | $CO_2$: 2013 - 2021 | 10.14291/tccon.ggg2020.edwards01.R0 |
| Lamont, OK | 36.604N, 97.486W | $CO_2$: 2008 - 2021 CO: 2017- 2021 | 10.14291/tccon.ggg2020.lamont01.R0 |
| Caltech, Pasadena, CA | 34.1362N, 118.126W | $CO_2$: 2012 - 2021 CO: 2017 - 2021 | 10.14291/tccon.ggg2020.pasadena01.R0 |
| East Trout Lake, Sask., CA | 54.354 N, 104.987W | $CO_2$: 2017 - 2021 CO: 2017 - 2021 | 10.14291/tccon.ggg2020.easttroutlake01.R0 |


**Table 1.** Location, dates of measurement, and DOIs of the TCCON sites used in this work. CO measurements require an InSb detector to cover the 2160 and 2111 $cm^{-1}$ windows, which has only been available since 2017 at Caltech, Lamont, and East Trout Lake.

## 2.2 The TARDISS Algorithm

Traditional profile retrievals fit spectra by adjusting the abundance of the trace gases at multiple vertical levels to determine the vertical distribution of a specific atmospheric species. Here, we focus on developing an algorithm that we are calling the Temporal Atmospheric Retrieval Determining Information from Secondary Scaling (TARDISS), that optimizes separating the profile of our target trace gas into two layers, one near the surface and the other at and above the middle troposphere (two 'partial columns', here scaled prior DMFs).

We use the notation and concepts of Rodgers and Connor (2003) with vectors represented with bolded lower-case letters and matrices represented with bolded upper-case letters. The following equations are used for each spectral window, each TCCON measurement, and each species retrieved (CO and $CO_2$ in this work) in the P1 fit. Equations 1 through 7 are used to calculate the weights and values that are used in Equation 8 and beyond as we shift the focus from one measurement of one spectral window to including all the measurements in a day of all the spectral windows used for a particular species. We will therefore keep the equations species and window agnostic for this description. We start with an equation expressing the calculation of the total column value:

$$\hat{z}_{P1} = z_{a,P1} + \boldsymbol{a}_{P1}^T \left( \boldsymbol{x}_{part} - \boldsymbol{x}_{a,P1} \right) \tag{1}$$





Where $\hat{z}_{P1}$ is the total column DMF output of a chosen species in a particular window from the P1 fit, $z_{a,P1}$ is the original vertical column DMF calculated from the prior profile scaled by the median VSF of the windows used, $\boldsymbol{a}_{P1}$ is the vector of column averaging kernel values output from the P1 TCCON processing weighted by the pressure thickness of each atmospheric layer, $\boldsymbol{x}_{a,P1}$ is the prior profile for the
chosen species also scaled by the median VSF of the windows used, and $\boldsymbol{x}_{part}$ is partial column DMF to be retrieved for the chosen species. All components in Equation 1 are in dry mole fractions and the averaging kernel is unitless. Equation 1 tells us how the retrieved DMF would change if the profile constructed from the two partial colummns differed from $x_{a,P1}$.

Focusing on the rightmost term of Equation 1, the averaging kernel is multiplied by the difference
of the prior and unknown DMF profiles summed for each level of the atmosphere.

$$\hat{z}_{P1} - z_{a,P1} = \boldsymbol{a}_{P1}^T \left(\boldsymbol{x}_{part} - \boldsymbol{x}_{a,P1}\right) = \sum_{i=1}^n a_{P1,i}(x_{part,i} - x_{a,i}) \tag{2}$$

Our method splits the total column at a specified altitude, q, and scales the prior profile below and above q independently such that:

$$\hat{z}_{P1} - z_{a,P1} = \sum_{i=1}^q a_{P1,i}(\gamma_L x_{a,P1,i} - x_{a,P1,i}) + \sum_{i=q+1}^n a_{P1,i}(\gamma_U x_{a,P1,i} - x_{a,P1,i}) \tag{3}$$

where $\gamma_L$ and $\gamma_U$ are the lower partial column scaling factor and upper partial column scaling factor, respectively. As this is linear, we group terms reducing the right side of Equation 3 to:

$$\hat{z}_{P1} - z_{a,P1} = (\gamma_L - 1)\sum_{i=1}^q a_{P1,i}\, x_{a,i} + (\gamma_U - 1)\sum_{i=q+1}^n a_{P1,i}\, x_{a,i} \tag{4}$$

Or further to:

$$\hat{z}_{P1} - z_{a,P1} = (\gamma_L - 1)J_L + (\gamma_U - 1)J_U \tag{5}$$

Where,

$$J_L = AW \sum_{i=1}^q a_{P1,i}\, x_{a,P1,i} \tag{6}$$

and

$$J_U = AW \sum_{i=q+1}^n a_{P1,i}\, x_{a,P1,i} \tag{7}$$

$J_L$ and $J_U$ both reduce to scalar values for each spectral window and prior profile. The AW term is a weighting term based on daily anomaly values between individual windows and is referred to as window





weighting from here on. We discuss the choices and reasoning for the AW term in Sect. 3.1.1 and Appendix A.

Equation 5 is applicable to all spectral windows for each spectrum measured. For example, for our $CO_2$ retrieval we use four separate spectral windows per measured spectrum and often have a few hundred spectra measured per day.

Our TARDISS retrieval uses an entire day's worth of TCCON retrievals in order to utilize the information in the temporal dimension. We combine the above equations into a matrix form:

$$y = Kx_\gamma + \epsilon \tag{8}$$

Where $y$ is a vector of values from the left side of Equation 5 for the $\alpha$ number of windows and k number of spectra over a day,

$$y = \begin{bmatrix} (\hat{z}_{P1} - z_{a,P1})_{w1,1} \\ \vdots \\ (\hat{z}_{P1} - z_{a,P1})_{w1,k} \\ \vdots \\ (\hat{z}_{P1} - z_{a,P1})_{w\alpha,1} \\ \vdots \\ (\hat{z}_{P1} - z_{a,P1})_{w\alpha,k} \end{bmatrix} \tag{9}$$

$K$ is the matrix of the $J_L$ and $J_U$ values for the $\alpha$ number of windows and k number of spectra over a day,

$$K = \begin{bmatrix} J_{L,w1,1} & & 0 & J_{U,w1,1} & & 0 \\ & \ddots & & & \ddots & \\ 0 & & J_{L,w1,k} & 0 & & J_{U,w1,k} \\ \vdots & \vdots & \vdots & \vdots & \vdots & \vdots \\ J_{L,w\alpha,1} & & 0 & J_{U,w\alpha,1} & & 0 \\ & \ddots & & & \ddots & \\ 0 & & J_{L,w\alpha,k} & 0 & & J_{U,w\alpha,k} \end{bmatrix} \tag{10}$$



and $x_\gamma$ is our state vector of partial column scalars which are the same for all windows in each measured spectrum.

$$x_\gamma = \begin{bmatrix} (\gamma_L - 1)_1 \\ \vdots \\ (\gamma_L - 1)_k \\ (\gamma_U - 1)_1 \\ \vdots \\ (\gamma_U - 1)_k \end{bmatrix} \qquad (11)$$

With $k$ measurements made in a day, four spectral windows, and two partial columns, the $y$ vector is of the size 4k by 1, the $x_\gamma$ vector is of the size 2k by 1, and the $K$ matrix is of the size 4k by 2k.

Fitting over an entire day of TCCON retrievals reduces the retrieved partial column error values compared to fitting individual measurements using Equation 5. Appendix B shows the influence of including multiple observations on the retrieved partial column errors.

       We use the maximum a posteriori (MAP) approach (Rodgers 2008) to calculate the most probable state vector from the given models and prior information. In line with the assumptions of the MAP approach, we assume our problem is linear and follows a gaussian distribution. The MAP solution can

take a few equivalent forms. In this work we use:

$$\widehat{x}_\gamma = x_{a,\gamma} + S_a K^T (K S_a K^T + S_\epsilon)^{-1}(y - K x_{a,\gamma}) \qquad (12)$$

Where $x_{a,\gamma}$ is the prior partial column scalar values, $S_a$ is the prior covariance matrix, $K$ is the forward mapping matrix, $S_\epsilon$ is the model covariance matrix, $y$ is the measurement vector, and $\widehat{x}_\gamma$ is the output

solution vector. The input components ($x_{a,y}$, $S_a$, and $S_\epsilon$) are described in Sect. 2.3.2.

       Once we have calculated the most likely solution for the partial column scalars, $\widehat{x}_\gamma$, we reconstruct the partial column DMF for the lower and upper partial columns as:

$$x_{PC} = \widehat{x}_\gamma x_{a,P1} + x_{a,P1} \qquad (13)$$



where $\boldsymbol{x}_{a,P1}$ is the prior partial column DMF calculated by integrating the median P1 posterior profile
using the same method as the standard TCCON full column retrievals (Wunch et al., 2011).

The MAP retrieval allows us to calculate the information content of the retrieval. In particular, we compare the degrees of freedom for our retrieval calculated by taking the trace of the averaging kernel of the fit, calculated as the following:

$$DoFs = tr(\boldsymbol{A}) = tr((\boldsymbol{K}^T \boldsymbol{S}_\epsilon^{-1} \boldsymbol{K} + \boldsymbol{S}_a^{-1})^{-1} \boldsymbol{K}^T \boldsymbol{S}_\epsilon^{-1} \boldsymbol{K}) \tag{14}$$


as well as the Shannon information content derived from the natural log of the determinant of the difference between the averaging kernel and an identity matrix:

$$H = -\frac{1}{2} \ln (|\boldsymbol{I} - \boldsymbol{A}|) \tag{15}$$

Generally, profile retrieval averaging kernels represent the sensitivity of a specific level of a profile to the rest of the levels in the profile. The averaging kernel for the TARDISS inversion is primarily a temporal averaging kernel relating how each partial column calculation relates to every other measurement during a day. We normalize the degrees of freedom by the number of measurements in each day for a more comparative understanding of the TARDISS degrees of freedom with respect to a 300   traditional profile retrieval as well as between days with a large variation in the number of measurements.

In order to compare the partial column retrievals to the in situ profiles for validation purposes, we calculate the vertical sensitivities of the TARDISS fit (shown in Fig. 4) using the gain matrix, **G**, from the TARDISS inversion and the averaging kernel profiles from the P1 measurement windows as:

$$\boldsymbol{G} = (\boldsymbol{K}^T \boldsymbol{S}_\epsilon^{-1} \boldsymbol{K} + \boldsymbol{S}_a^{-1})^{-1} \boldsymbol{K}^T \boldsymbol{S}_\epsilon^{-1} \tag{16}$$


$$\boldsymbol{A}_{vert} = \boldsymbol{x}_{a,P1} * \boldsymbol{G} * \boldsymbol{a}_{P1} \tag{17}$$

where $\boldsymbol{a}_{P1}$ is the same vector of column averaging kernels from Equation 1.

Since $\boldsymbol{a}_{P1}$ represents the change in TCCON $X_{gas}$ DMF per change in true DMF at each level $(\frac{\delta X_{gas,TCCON}}{\delta x_{true}})$ and the gain matrix represents the change in partial column scalar per change in TCCON 310   $X_{gas}$ DMF $(\frac{\delta \gamma}{\delta X_{gas,TCCON}})$, $\boldsymbol{A}_{vert}$ has units of change in partial column scalar per change in level DMF





value ($\frac{\delta\gamma}{\delta x_{true}}$) and relies on the difference between a 'true' in situ profile and the prior profile used in the inversion.

### 2.3 Algorithm Setup and Choices

### 2.3.1 Pre-processing of the Phase 1 Data

We begin by preprocessing the P1 fits. We take the P1 model prior profile and scale it by the median value of the P1 output scalar values for each spectrum from the windows used so that our TARDISS fit is centered around the median P1 posterior profile for each measurement point. This assumes that the true column VMR of a species is some linear combination of the VMRs calculated from the windows used in the TARDISS fit. Then, we calculate the partial column priors by integrating

the scaled prior profiles over the respective pressure levels for each chosen partial column. Finally, we assemble the necessary matrices for the fit described by Equation 12.

### 2.3.2 Maximum a Posteriori Components

        The different components of Equation 12 reflect where prior information can be used in the algorithm and several additional choices can be made to improve the fit. The following describes our

standard input for these components. We present tests of the retrieval's sensitivity to these choices in Sect. 3.1.1.

        For the prior covariance matrix, we use an identity matrix for the lower partial column scalar portion of the covariance matrix, and we use an exponential decay from the diagonal for the upper partial column scalar portion of the covariance matrix. This requires that upper column scalar values

shift in relation to one another and theoretically imposes higher costs if the upper partial column scalars change rapidly in time. The off-diagonal values of the upper partial column portion of the prior covariance matrix decay with respect to the measurements made before and after them over the course of one-third of a day of measurement. Since the prior covariance matrix is inverted in the calculations, decreasing the magnitude of the prior covariance matrix scalar increases the constraints imposed during

the calculations so that a scalar of $10^{-5}$ is a larger constraint than a scalar of $10^{-4}$. A discussion of the influence of the temporal covariance is in Sect. 3.1.3.





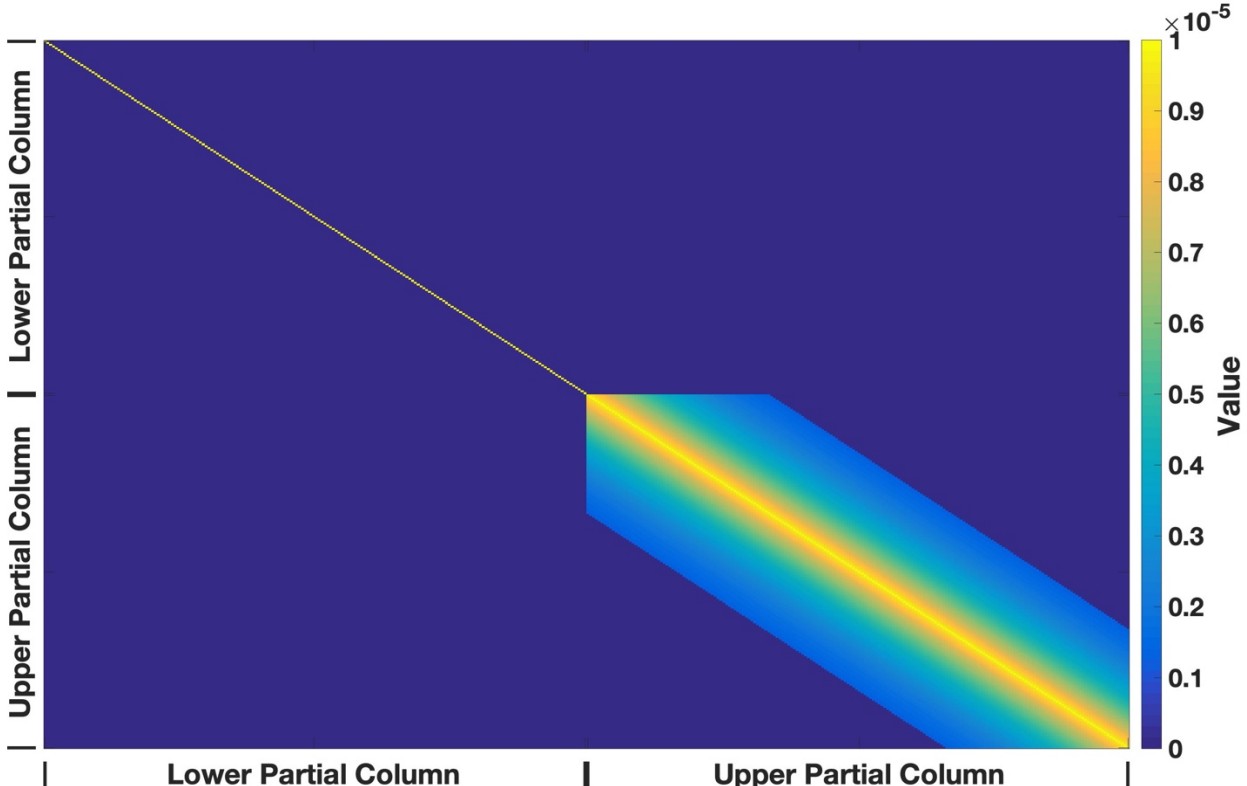

**Figure 2.** Example of a prior covariance matrix color coded by the magnitude of the value. The lower partial column has a prior covariance that is a scaled identity matrix, the upper partial column has a prior covariance that decays over one third of the measurement day, and the cross covariances between the upper and lower partial columns is assumed to be zero.

The measurement error covariance matrix is a diagonal matrix composed of the squares of the P1 errors for each spectral window so that measurements with smaller errors are weighted more heavily than those with larger errors.

$CO_2$ and CO use different values for the prior TARDISS scale factors ($x_{a,\gamma}$). For CO, we assume a uniform prior scale factor of one for all observations. For $CO_2$ we solve Equation 12 using the linear least-squares method:

$$x_{L2} = (K^T K)^{-1} K^T y \qquad (18)$$

and use the median daily value of $x_{L2}$ as $x_{a,\gamma}$ in Equation 12. While the linear least-squares method provides a solution to our retrieval, it does not allow us to utilize additional prior information in the


covariance of the partial columns or to specify prior partial column scale factors. Including these pieces of information reduces the retrieved partial column error values as shown in Fig. B. in Appendix B.

We adopted different approaches for these two gases since using a static prior of one for the $CO_2$ retrievals worsened the comparison to in situ data but improved the validation comparison for the CO

retrievals (shown in Sect. 3.1.1).

### 2.3.3 Choosing Spectral Windows for the TARDISS Fit

The primary information content used in our algorithm is derived from the fact that the total column abundances retrieved from different spectral windows of the same species will differ due to differences in their kernels, unless the shape of the a priori profile is perfect. Thus, differences in the

retrieved columns from different windows, together with their kernels, can be used to infer the errors in the a priori VMR profile, and hence derive a better VMR profile than one which is determined by simply scaling the a priori VMR profile. For this method to have sufficient information, windows with different vertical averaging kernels are needed, such as those shown in Fig. 1. Preferably, the windows used for the TARDISS retrieval would have a window that is more sensitive to the lower atmosphere

and a window that is more sensitive to the upper atmosphere so that a larger amount of information is contained between them. While it is imperative to use windows that have differing averaging kernel profiles, it is also necessary to use windows that have low error in the P1 fit. The higher the error in a particular spectral window, the more uncertainty that that retrieval will add to the TARDISS results.

For the partial column $CO_2$ calculations, we use four spectral windows in the P1 process

centered at 6339, 6220, 4852, and 6073 cm$^{-1}$. The 6339 cm$^{-1}$ and 6220 cm$^{-1}$ windows are spectroscopically similar and have column averaging kernel profiles that vary with solar zenith angle providing some vertical information over the course of a day. The 4852 cm$^{-1}$ window has an averaging kernel profile that is largest at the surface and minimal at upper troposphere and lower stratosphere and the 6073 cm$^{-1}$ window has an averaging kernel profile that is effectively the opposite of the 4852 cm$^{-1}$

window. Both the 4852 cm$^{-1}$ and 6073 cm$^{-1}$ window averaging kernels are largely independent of solar zenith angle with the exception of the highest levels in the 6073 cm$^{-1}$ window profile.

For the partial column CO calculations, we use three spectral windows fit during the P1 process. There is one window in the NIR region centered at 4233 cm$^{-1}$ and two windows in the MIR region centered at 2111 and 2160 cm$^{-1}$. The two MIR windows have similar averaging kernel profiles that

maximize at the surface and drop to nearly zero at upper levels. The NIR window averaging kernel profile has a minimum at the surface and a maximum at the upper levels.

Unlike the $CO_2$ windows that are all observed by the InGaAs detector, the MIR CO windows are measured by a liquid nitrogen cooled InSb detector. For this reason, we only have results of the CO partial column fits at the Caltech, Lamont, and East Trout Lake TCCON sites and, due to the lack of in





situ profiling data in Pasadena, we only have direct vertical profile comparison results from the Lamont
and East Trout Lake TCCON site.

Other windows measured by TCCON instruments were considered for the partial column
calculations for both species however they had high levels of error in the P1 fit, had fits that were
particularly sensitive to changes in temperature, or their averaging kernels were similar enough to the
other windows that they would not provide any extra information while still increasing the error.

### 2.3.4 Choice of Partial Column Height

We chose the lower partial column to integrate from the surface through the first five vertical
layers of the GEOS meteorological fields. Using this criterion, a site at sea level has a lower column
from sea level to 2 km and the upper partial column from 2 to 70 km. While somewhat arbitrary, the
choice of 2 km was made to have the lower partial column encompass the surface mixed layer at most
locations while minimizing the dilution of surface exchange signals that would result from integrating
over a larger partial column. If there are known enhancements of significant species enhancement near
the 2 km level (such as CO during wildfire season), the retrieval performance may be degraded and a
different partial column height may be a more appropriate choice.

## 2.4 Comparison Data

### 2.4.1 In situ Vertical Profile Data

We use data from multiple aircraft and AirCore campaigns between 2008 and 2020 (Cooperative
Global Atmospheric Data Integration Project; (2019)) to evaluate our partial column retrieval. All in
situ $CO_2$ data are reported on the WMO X2007 scale. The aircraft data are from the SEAC4RS
campaign in 2013, the ATom and KORUS-AQ campaigns in 2016, and from measurements made by
the Goddard Space Flight Center between 2014 and 2016. We use AirCore measurements taken in July
of 2018 at the Armstrong, Lamont, and Park Falls sites. In addition, we use measurements from the
NOAA Global Greenhouse Gas Reference Network's Aircraft Program to compare with our lower
column measurements. Table S2 provides a summary of the in situ data used in this work.

The NASA Studies of Emissions and Atmospheric Composition, Clouds and Climate Coupling
by Regional Surveys (SEAC4RS) campaign used an AVOCET instrument on the ER-2 aircraft to
sample in situ $CO_2$ in the atmosphere from the surface into the lower stratosphere (Toon et al., 2016).
The SEAC4RS measurements align with TCCON measurements at the Armstrong site on 23 September
2013 where the ER-2 sampled from 1.5 km to 19 km altitude.

The Korea-United States Air Quality Study (KORUS-AQ) campaign used a multitude of
instruments and platforms to measure and better understand air quality in South Korea and how to
improve it (Crawford et al., 2021). One of the measurement platforms was the NASA DC-8 aircraft





which made coincident measurements with the TCCON instrument at the Armstrong site on 18 June
2016. Measurements of $CO_2$ using a non-dispersive IR spectrometer during a flight that sampled from
0.68 km to 12 km are used here for comparison to retrieved partial columns.

The Atmospheric Tomography Mission (ATom) generated a global dataset to study the
interactions of anthropogenic air pollution and greenhouse gases from Summer 2016 through Spring
2018 (Wofsy et al., 2021; Thompson et al., 2022). During this mission, the NASA DC-8 aircraft made
measurements of trace gases worldwide. On 22 August 2016, their path coincided with the TCCON
measurements at the Park Falls site, where they measured in situ $CO_2$ using a Picarro cavity ringdown
spectroscopy (CRDS) trace gas analyzer (Crosson, 2008) from 0.79 km to 12 km.

We use Goddard Space Flight Center aircraft measurements that were made at the Armstrong
site on 20 and 22 August 2014, 02 October 2015, and 10 February 2016. Picarro CRDS measurements
were made on these flights up to ~13 km and down to 0.6 km. Multiple measurements were made on 20
and 22 August 2014.

The AirCore sampling system is composed of coiled stainless-steel tubing that is open on one
end and samples ambient air as it descends from a balloon flight. This sample is then analyzed using a
Picarro CRDS trace gas analyzer using an algorithm that accounts for the effect of longitudinal mixing
on vertical resolution (Karion et al., 2010; Tans, 2009). We use AirCore data from measurements made
on 16, 17, and 18 July 2018 at the Armstrong site; on 23, 25, and 27 July 2018 at the Lamont site, and
on 30 and 31 July 2018 at the Park Falls site.

Finally, we use CO and $CO_2$ data measured at the Lamont site (site code SGP) and at the East
Trout Lake site (site code ETL) as a part of the NOAA Global Greenhouse Gas Reference Network's
Aircraft Program to measure the seasonal climatology of greenhouse gases in North America (Sweeney
et al., 2015). We use data from 282 of the 399 flights made between 2008 and 2018 and all 34 flights
for CO made between 2017 and 2020 at the SGP site. At the ETL site we use 26 of the 40 flights for
$CO_2$ and 26 of the 39 flights for CO made between 2017 and 2021. Most of the measurements were
made below 6 km altitude at the SGP site and below 7 km altitude at the ETL site, with some
measurements as low as 0.17 km altitude. We use data from individual flights that have more than 2
measurements points within our 1.9 km lower partial column and at least one point at or below 1 km.
Since these datasets do not include much data within the upper partial column, we compare with these
measurements only to our retrieved lower partial column values and exclude them from the validation
discussion in Section 3.1.

**2.4.2 Ground-based Data**

In Section 3.4, we use ground-based data to compare to our lower partial column retrieval output
and to begin discussions of the applications of our retrievals.





Directly next to the Park Falls TCCON site is a 400 m tall tower that is part of the NOAA GGGRN tall-tower and AmeriFlux networks and has multiple vertical levels altitudes along the tower at which measurements are made (Berger et al., 2001; Andrews et al., 2014). The tall tower uses a Licor LI-6262 to measure $CO_2$, and we use the measurements at the 396 m level to compare to our lower partial column retrievals.

A Cimel CE-318-N multiband photometer measures aerosol optical depth as a part of the AERONET network from the top of the Caltech Hall roughly 100 m from the Caltech TCCON site (Holben et al., 1998). It measures aerosol optical depth at 340, 380, 440, 500, 675, 870, and 1020 nm as well as atmospheric water vapor in centimeters. The photometer measures in 15-minute intervals and has been measuring at this location since 2010. We compare the level 1.5 cloud screened aerosol measurements to our lower partial column CO measurements to explore the connection between emissions, meteorology, and aerosols in Sect. 3.4.

## 3 Results and Discussion

To understand the effectiveness of our partial column retrieval, we use an adjusted version of the smoothing calculation shown in Equation 3 of Wunch et al. (2010) to determine the value the inversion would return if it were using the true profile instead of the scaled P1 priors:

$$\hat{z}_s = z_{a,P1} + \boldsymbol{A}_{vert}\left(\boldsymbol{x}_{true} - \boldsymbol{x}_{a,P1}\right) \tag{19}$$

where $\boldsymbol{x}_a$ is the prior profile used in Equation 1 and $\boldsymbol{x}_{true}$ is the measured in situ profile converted to DMF. The in situ profile is interpolated to the same vertical levels as the TCCON prior profile as shown in Fig. 3. After calculating the smoothed in situ profile, we integrate from the surface of the profile to the same vertical level at which the partial column is separated, q in Equation 3, for the lower column or that level to the top of the atmosphere for the upper column using the method outlined in Wunch et al. (2010). We then compare the integrated, smoothed, in situ partial column DMF directly to the output from the reconstructed lower and upper partial columns calculated in Equation 13.





**Figure 3.** An example of the profiles used in the direct comparison calculations using data from the Park Falls site on July 27, 2018. The profile above 6 km is not shown. The solid black line is the TARDISS prior profile scaled by the median of the TCCON vertical scaling factors from the spectral windows used. The green dot-dashed line is the measured AirCore mixing ratios. The red, dashed line is the AirCore measurements interpolated to the vertical spacing of the TARDISS prior, and the blue, dotted line with circles is the smoothed, vertical sensitivity weighted profile that is integrated to calculate the partial column that our phase 2 fitting would calculate if it had a 'true,' AirCore profile. The black dots within the blue circles represent the points of the profile that make up the lower partial column.





The TARDISS algorithm is very efficient – taking only a minute of processing time per year of data for each species. This efficiency enables the validation comparisons to be performed using many different model choices. Thus, we evaluated the sensitivity of the TARDISS inversion by varying different forward model choices. The set of choices that we have designated as the operational setup for $CO_2$ inversion are:

• The covariance matrix is scaled by $10^{-5}$ to better constrain the fit
   • The prior scalar for the lower and upper partial column scalar is the daily median of the least squares solution for the respective column
   • The J values for each window are weighted by the square of the respective daily anomaly ratio of the individual columns (further explained in Appendix A).

For the CO inversion, the operational setup parameters are:
   • A covariance matrix scaled by $10^{-5}$
   • An ideal prior partial column scalar of one
   • The J values are left unweighted.

The daily anomaly ratios weightings are only applied to the $CO_2$ fits as changing the weightings in the CO inversion did not significantly affect the output. We vary three aspects of the algorithm and observe the differences in the validation comparisons. The results of these tests are discussed in Sect. 3.1.1 and represented in Table 2 and Table 3.

**3.1 Validation Comparisons**

   We compare retrieved partial column values from three of the five sites presented in this work using measurements from the same set of in situ data used to evaluate and derive the so-called 'in situ scaling factor' of the P1 TCCON retrievals. For $CO_2$, there are twenty-four points of comparison obtained between 2013 to 2018. Twelve of those comparisons are from the Armstrong TCCON site
spanning 2013 to 2018 and during different months of the year. Four profiles are available above the Park Falls TCCON site (August 2016 and July 2018). The remaining eight profiles are from the Lamont TCCON site and are all from July 2018. As the Lamont site is the only site in this work with an InSb detector and overlapping in situ measurements, the eight profiles measured at the Lamont site serve as the entire CO comparison.
The comparison profiles were measured by aircraft-based instruments or AirCore measurements, all described in Sect. 2.4.1. We compare the TARDISS retrievals from spectra obtained within one hour of the of the in situ profile. One benefit of the $A_{vert}$ term in Equation 19 is that it includes the temporal sensitivity of the inversion data which is not able to be taken into account in other remote sensing validations; however, the temporal sensitivity (represented by off diagonal terms in $A_{vert}$) is often close





to zero so that the retrieved partial column values have little influence on the data before or after them. We report linear fits between the partial column retrievals and the integrated, smoothed, in situ partial columns with y-intercepts forced through zero. Since our retrieval is designed to be linear, we use fits with y-intercepts forced through zero. As there are only scaling values in our retrieval, a non-zero y-intercept would introduce spurious error into our analysis. Since the reported coefficient of

determination (commonly referred to as the $r^2$ value) for this linear fit would be spuriously high, we report the mean ratio of our retrieved partial column to the integrated, smoothed, in situ measurement as it deviates from one. This mean ratio deviation value gives a more direct understanding of how the partial column values compare as a lower value signifies a better comparison.

We use these validation comparisons to perform sensitivity tests of our algorithm parameters

and determine an operational set of parameters. We then use these optimal parameters for the $CO_2$ and CO retrievals to quantify the total error of our retrieval by calculating a validation error multiplier for each site.

### 3.1.1 Sensitivity Analysis

Several terms in our retrieval do not have a single, unambiguously correct choice for their

values. To evaluate the sensitivity our retrieval to the choices made for these parameters, we have run our retrieval with alternate choices and report the degrees of freedom and comparison to in situ data (specifically, the retrieval comparison error, slope of the zero-forced linear fit, and the mean ratio deviation value of the linear fit) for each test. We tested changes to three terms: the TARDISS scale factor priors, the a priori covariance matrix scaling, and changes to the weightings on the individual

windows.

To test the sensitivity of the retrieval to the partial column scalar prior, we compare the changes in the validation when using $\boldsymbol{x}_{L2}$ from Equation 18 as the prior, the daily median of $\boldsymbol{x}_{L2}$ (our operational choice for $CO_2$), as well as the idealized scalar of unity (our operational choice for CO) to each other. In Tables 2 and 3, these are identified as "Xl2," "Xl2 daily median," and "static ideal prior," respectively.

We also test the sensitivity of the retrieval to how the prior covariance matrix is scaled. Doing so changes how strongly the retrieval is constrained to the prior. Here, we alter the a priori covariance matrix by scaling the matrix uniformly to evaluate the optimal weighting with respect to the validation comparisons. We report the results of using scaling values of $1x10^{-4}$, $5x10^{-5}$, and $1x10^{-5}$. While other scaling values were tested, the resulting errors were large enough or the resulting degrees of freedom

were small enough, that the values were disregarded from further study.

Finally, we test the sensitivity to scaling the window weighting matrices defined by Equations 6 and 7. In our testing, we scale these matrices (the AW term in Equation 6 and 7) by one, the daily anomaly ratios discussed in Appendix A, and the square of the daily anomaly ratios. We multiply the matrices by these scalars since each spectral window is sensitive to different parts of the atmosphere and therefore





returns a different total column $X_{gas}$ value for the same atmospheric state. The ratio of the daily
       anomalies measured with the different windows used allows us to weight the windows in the inversion
       based on their observed relationships. Scaling the window weightings for the $CO_2$ retrieval by the
       square of the daily anomaly ratios gave the best comparison likely due to the increase of the
       contribution of the 4852 cm$^{-1}$ window to the retrieval. Scaling the window weightings for the CO
retrieval had little effect so we proceed with results retrieved with unscaled window weightings. We
       show the results of the scaled $CO_2$ retrieval comparisons in Table 2 and the results of the unscaled CO
       retrieval comparisons in Table 3.
           For the $CO_2$ retrievals, the best performance and the operational set of parameters comes from using
       the daily median of the least squares solution as the prior, scaling the a priori covariance matrix by $10^{-5}$,
and scaling the window weights by the square of the daily anomaly ratios. The degrees of freedom per
       measurement changes with the scaling of the prior covariance matrix and varies between 0.364 (16.1 for
       the overall comparison) for the $10^{-5}$ scaling increasing to 0.800 (34.5 overall) for the $10^{-4}$ scaling. The
       same pattern holds for the errors with values of 0.586 ppm and 0.942 ppm for the $10^{-5}$ scaling increasing
       to 1.331 ppm and 1.661 ppm for the $10^{-4}$ scaling for the upper and lower columns, respectively.
The validation slopes change with both the prior covariance matrix scaling and the prior scalar
       choice. As we are trying to determine the parameters that give the best comparison results between the
       in situ and lower partial column $CO_2$ data specifically, we chose the parameters that resulted in the
       validation slope closest to one for the lower partial column. The validation slope closest to one is 1.007
       which is the results of using the daily median of the $x_{L2}$ values as a prior and scaling the prior
covariance matrix by $10^{-5}$. The validation slope for the upper column comparison with these parameters
       is 1.004 which is comparable to values from other parameter choices. While scaling the prior covariance
       matrix by $5\times10^{-5}$ and using a prior scalar of unity optimizes the upper partial column comparison, it
       leads to a poorer lower column comparison which is counter to the intention of the algorithm. The full
       comparison data between retrieval parameters is shown in Table 2 and the comparison data for the
unscaled window weights are shown in Table S1. The poor lower column comparisons and excellent
       upper column comparisons for the retrievals using no window weightings shown in Table S1 suggest
       that the window weightings are acting to emphasize the information from the 4852 cm$^{-1}$ window (and
       theoretically the surface) as intended. Since the largest variation in validation slopes in either column
       and for different window weightings is driven by the change in the prior partial column scalar, we posit
that the prior partial column scalar choice is the most significant parameter in the $CO_2$ retrieval.



| TARDISS Prior Choice | Prior Covariance Matrix Scaling | DoF per measurement (overall) | Lower Column Error (ppm) | Lower Column Validation Slope | Lower Column Mean Ratio Deviation | Upper Column Error (ppm) | Upper Column Validation Slope | Upper Column Mean Ratio Deviation |
|---|---|---|---|---|---|---|---|---|
| Xl2 daily median | $10^{-5}$* | 0.364 (16.1) | 0.942 | 1.007 | 0.010 | 0.586 | 1.004 | 0.012 |
| | $10^{-4}$ | 0.800 (34.5) | 1.661 | 1.012 | 0.014 | 1.331 | 1.005 | 0.012 |
| | $5 \times 10^{-5}$ | 0.700 (30.4) | 1.444 | 1.010 | 0.012 | 1.042 | 1.005 | 0.012 |
| Xl2 | $10^{-5}$ | 0.364 (16.1) | 0.942 | 1.008 | 0.010 | 0.586 | 1.004 | 0.012 |
| | $10^{-4}$ | 0.800 (34.5) | 1.661 | 1.012 | 0.014 | 1.331 | 1.004 | 0.011 |
| | $5 \times 10^{-5}$ | 0.700 (30.4) | 1.444 | 1.010 | 0.012 | 1.042 | 1.004 | 0.012 |
| Static ideal prior | $10^{-5}$ | 0.364 (16.1) | 0.942 | 1.013 | 0.013 | 0.586 | 0.998 | 0.005 |
| | $10^{-4}$ | 0.800 (34.5) | 1.661 | 1.016 | 0.016 | 1.331 | 1.001 | 0.007 |
| | $5 \times 10^{-5}$ | 0.700 (30.4) | 1.444 | 1.014 | 0.015 | 1.042 | 1.000 | 0.006 |

**Table 2.** Variations in $CO_2$ retrieval upper and lower column validation slopes, upper and lower column mean ratio deviations, upper and lower column comparison errors, and DoF for different TARDISS prior choices and prior covariance matrix scaling values. All values shown here are for retrievals using the square of the anomaly weightings. The asterisk in the first row indicates that this is the operational set of parameter choices for the $CO_2$ retrieval.


For the CO retrievals, the best performance uses a static prior scalar of one, scaling the prior covariance matrix by $10^{-5}$, and not scaling the window weights. The degrees of freedom per measurement increases from 0.170 (6.77 overall) to 0.638 (25.3 overall) and the errors increase from 0.174 ppb and 0.413 ppb to 0.402 ppb and 0.861 ppb as we change the scaling of the prior covariance
matrix from $10^{-5}$ to $10^{-4}$ for the upper and lower columns, respectively. All the tests resulted in similar values of mean ratio deviation and errors under 1 ppb but using a static prior scalar of one and scaling the prior covariance matrix by $10^{-5}$ resulted in the lower column validation slope that was closest to one (1.001), so we chose this set of parameters as the operational parameters for the CO retrievals. These parameters also had an upper column validation slope of 1.031 which is closer to one than slopes from
most other parameter choices.



| TARDISS Prior Choice | Prior Covariance Matrix Scaling | DoF per measurement (overall) | Lower Column Error (ppb) | Lower Column Validation Slope | Lower Column Mean Ratio Deviation | Upper Column Error (ppb) | Upper Column Validation Slope | Upper Column Mean Ratio Deviation |
|---|---|---|---|---|---|---|---|---|
| Xl2 daily median | $10^{-5}$ | 0.170 (6.77) | 0.413 | 0.977 | 0.024 | 0.174 | 1.079 | 0.078 |
| | $10^{-4}$ | 0.638 (25.3) | 0.861 | 0.987 | 0.014 | 0.402 | 1.095 | 0.094 |
| | $5\times10^{-5}$ | 0.485 (19.3) | 0.727 | 0.983 | 0.017 | 0.310 | 1.092 | 0.091 |
| Xl2 | $10^{-5}$ | 0.170 (6.77) | 0.413 | 0.973 | 0.028 | 0.174 | 1.088 | 0.087 |
| | $10^{-4}$ | 0.638 (25.3) | 0.861 | 0.985 | 0.017 | 0.402 | 1.101 | 0.100 |
| | $5\times10^{-5}$ | 0.485 (19.3) | 0.727 | 0.980 | 0.021 | 0.310 | 1.098 | 0.098 |
| Static ideal prior | $10^{-5}$ * | 0.170 (6.77) | 0.413 | 1.001 | 0.005 | 0.174 | 1.031 | 0.033 |
| | $10^{-4}$ | 0.638 (25.3) | 0.861 | 1.007 | 0.012 | 0.402 | 1.053 | 0.055 |
| | $5\times10^{-5}$ | 0.485 (19.3) | 0.727 | 1.005 | 0.009 | 0.310 | 1.046 | 0.048 |

**Table 3.** Variations in CO retrieval upper and lower column validation slopes, upper and lower column mean ratio values, upper and lower column comparison errors, and DoF for different TARDISS prior choices and prior covariance matrix scaling values. The asterisk in the second to last row indicates that this is the operational set of parameter choices for the CO retrieval.


### 3.1.2 Standard Output

Using the operational setup for our TARDISS fit, the comparison of $CO_2$ between our output lower partial column VMRs and the integrated, smoothed, in situ partial columns for all of the sites
gives a slope of 1.007, a best fit standard error of 0.002 on the slope, and a corresponding mean ratio deviation value of 0.010. Figure 4 shows the 1-to-1 comparison plots for the upper and lower partial columns for $CO_2$ and CO. Across all twenty-four points, the average degrees of freedom of the daily inversions was 0.364 degrees of freedom per measurement for the lower column and the comparisons include between thirteen and sixty-seven inverted partial column data points. The error in the lower
partial column values was 0.942 ppm on average across the comparison and the average reported error from the in situ profiles was between 0.03 and 0.156 ppm for the comparative partial column. The comparisons vary by site with the Lamont comparisons having the largest offset slope of 1.008. The Park Falls and Armstrong comparisons have slopes of 1.000 and 1.006 respectively. Since neither the





P1 or TARDISS retrieval can fully optimize the shape of the partial profile that they are scaling, the site
to site differences are likely due to the variation of ability of the TCCON P1 priors to capture the
source, sink, and transport complexities in the shape of the prior profile that directly translate to the
DMFs used in the partial column scaling.

The operational comparison for the lower partial column CO fit gives a slope of 1.001, a best fit
standard error of 0.002, and a corresponding mean ratio deviation value of 0.005. The inversion days
have an average of 0.170 degrees of freedom per measurement with between 33 and 43 data points per
comparison. The error in the lower partial column values was 0.413 ppb on average across the
comparison.

To quantify the total error of our retrieval, we use the 1-to-1 comparisons to scale our error
values to the point where at least 50% of the comparison points are within the error range of the 1-to-1
line. We calculate the scalar values as:

$$VEM = Median(\frac{|\hat{z}_{comp} - \hat{z}_s|}{\sigma}) \qquad (18)$$

where $\hat{z}_{comp}$ is the comparison partial column value, $\hat{z}_s$ is the integrated, in situ partial column value, $\sigma$
is the output total retrieval error, and VEM is the calculated validation error multiplier. For $CO_2$ at Park
Falls, the lower and upper column VEM are 1.23 and 11.7; at Armstrong, the lower and upper column
values are 4.48 and 8.33; and at Lamont the values are 3.43 and 6.29 for the lower and upper column,
respectively. Since Caltech and East Trout Lake do not have comparison data, we apply error multiplier
values of 4.48 and 11.7 as they are the largest multiplier values from among the other sites. For CO, the
Lamont site the multiplier values are 1.57 and 12.5 which we use for the Caltech and East Trout Lake
site CO retrieval data as well. The upper column VEM values are consistently larger than the associated
lower column values; however, since the upper column errors are smaller than the lower column errors,
the total errors for the lower and upper columns are closer in magnitude than the VEM values.



**Figure 4.** The direct comparisons between the partial column DMF values retrieved from the TARDISS fit and the integrated, smoothed aircraft partial columns for $CO_2$ (a,b) and the CO (c,d) for the lower (a,c) and upper (b,d) columns. The $CO_2$ comparisons are color coded by site and the CO comparisons are solely from the Lamont site. The error bars in the x-direction are the reported errors from the aircraft data smoothed the same way as the in situ measurements and the error bars in the y-direction are the output errors from the TARDISS fit. The black solid line is the 1-1 line and the blue line is the linear fit of the data with the y-intercept forced through zero.




### 3.1.3 Temporal Covariance

The analysis above implemented an a priori covariance matrix with temporal covariance between upper partial column scalars over the course of a day of measurement, shown in Fig. 2. To determine how this affects our retrievals, we compare the data above to the validation comparison from a $CO_2$ retrieval not constrained by a temporal covariance. The prior covariance matrix without the temporal covariance is simply a diagonal matrix of the $10^{-5}$ scalar value. Table 4 shows that the retrievals without temporal constraints have a slightly poorer validation comparison overall, including higher errors and fewer degrees of freedom. However, the site by site differences in validation data show that the lower column VEM for the Armstrong site is smaller when using a temporally unconstrained fit, whereas both the Park Falls and Lamont VEMs are improved when implementing the temporal constraints. Similarly, the upper column VEM values at all sites improve without the temporal constraints. While the purpose of this publication is to create an operational product, the varying effects of the prior covariance matrix choice on the site VEMs suggest that the site by site parameter choices could be individually determined in order to minimize error and increase the partial column precision if there were sufficient in situ validation profiles over the measurement site.

| Statistics | | Temporally Constrained Upper Column | Temporally Unconstrained Upper Column |
|---|---|---|---|
| Validation DoF (Overall) | | 0.364 (16.1) | 0.317 (13.9) |
| Lower Column $CO_2$ | | | |
| | Error (ppm) | 0.942 | 0.978 |
| | Validation Slope | 1.007 | 1.008 |
| | Mean Ratio Deviation | 0.010 | 0.012 |
| | Park Falls VEM | 1.23 | 1.52 |
| | Armstrong VEM | 4.48 | 4.35 |
| | Lamont VEM | 3.43 | 5.09 |
| Upper column $CO_2$ | | | |
| | Error (ppm) | 0.586 | 1.053 |
| | Validation Slope | 1.004 | 1.002 |
| | Mean Ratio Deviation | 0.012 | 0.013 |
| | Park Falls VEM | 11.7 | 9.04 |
| | Armstrong VEM | 8.33 | 5.26 |
| | Lamont VEM | 6.29 | 3.36 |

**Table 4.** Validation comparison DoF, error, validation slope and mean ratio deviation and site VEM values for lower and upper column $CO_2$ for retrievals using a temporally constrained upper column and





a temporally unconstrained upper column. The retrievals are performed with the operational parameters denoted by asterisks in Table 2.

     The way that the temporal covariance impacts our validation comparison is through the partial column vertical sensitivities described in Equation 17 via the gain matrix (Equation 16). To assess

the importance of our chosen prior covariance matrix, we compare the vertical sensitivities for a temporally constrained upper column and a temporally unconstrained upper column (shown in Fig. 5) for a representative day (July 27th, 2018 at the Lamont site) using the operational parameters denoted by the asterisk in Table 2.

     Without the temporal constraint, the upper column sensitivities are on the same order as the

lower column sensitivities with values between -0.05 and 0.12. For reference, a change of 1 ppm at a level with a sensitivity of 1 would result in a change in the partial column scalar of 0.025 and partial column scalars for $CO_2$ change on the order of a few percent $O(0.01)$ per measurement. The upper column sensitivity peaks around the 10 km level at low solar zenith angles and the peak moves toward the surface at higher solar zenith angles. The lower column sensitivities always peak

at the surface but the sensitivity increases at higher solar zenith angles.

     With the temporal constraint, the pattern of the peaks with respect to SZA remains similar but the upper column sensitivities are roughly five times the value and the lower column sensitivities are half the value as the temporally unconstrained values. The imposed temporal covariance constrains the upper column to vary together over the span of a measurement day so that a change at one level

in the column at one measurement point would also induce changes at other measurement points therefore increasing the vertical sensitivities in the upper column over the entire retrieval day. This constraint is also stringent enough that it propagates into the sensitivity of the lower column scalar. Since our goal is to retrieve a lower partial column it seems counterintuitive that using sensitivities with an order of magnitude difference provides a better validation comparison. However, for this

method we assume that we know the shape and behavior of the upper column fairly well and that most of the change occurs near the surface. Given these assumptions, constraining the upper column more heavily by introducing expected daily patterns through the a priori covariance matrix allows for the lower column retrieval to have improved comparisons with in situ data despite decreased vertical sensitivities.

While we test retrievals with and without temporal covariance, the choice of prior covariance matrix shape could be much more complex. Future study could include using model generated or back trajectory based temporal covariances to include outside information in the retrieval dynamically. For an operational retrieval product, we will include the temporal covariance in the prior covariance matrix as an operational parameter.






**Figure 5.** Vertical sensitivities of the lower partial column (a,c) and upper partial column (b,d) scalars color coded by solar zenith angle in degrees. The sensitivities calculated when using a temporally covariant prior covariance matrix are shown in (a) and (b) and when using a non-temporally covariant prior covariance matrix are shown in (c) and (d).

## 3.2 Errors and Information Content

### 3.2.1 Error Analysis

Using the information from the validation comparison, we can study the errors of the entire dataset from each of the five sites. The output of the retrieval is the partial column scalar and the error retrieved is the standard deviation of the partial column scalar calculated from the retrieval variance and represented as another scalar value. To convert our partial column scalar error to parts per million, we multiply the error scalar value by the prior partial column mixing ratio. Error varies from site to site due





to variations in the P1 total column errors that are input to the measurement covariance matrix and due
to how well the prior partial column DMF matches the, generally unknown, actual partial column DMF.
We report the total retrieval error, retrieval error components, and the error contribution from the
validation comparison measurements in Table 5.

Amongst all the sites, the error values range from 0.977 ppm to 1.068 ppm for lower column $CO_2$
and from 0.605 ppm to 0.737 ppm for the upper column $CO_2$ with the highest average error in Park
Falls and the lowest average error in Armstrong for both columns. For CO retrievals, the average total
retrieval error ranges from 0.391 ppb to 0.832 ppb for the lower column and 0.105 ppb to 0.492 ppb for
the upper column. In general, the errors vary minimally with season, but the Lamont site has a distinct
seasonality for both lower column CO and $CO_2$ retrievals with the highest errors are during the summer
perhaps due to differences between the true and prior profiles at this site during summer (Fig. S1). The
errors for $CO_2$ generally increase over time since the prior partial column that is being scaled is
increasing with continuing anthropogenic emissions even though the scalar values remain similar across
the dataset for both CO and $CO_2$ (Fig. S2).

The total retrieval error is the total of the model parameter error, the smoothing error, and the
retrieval noise (Rodgers, 2008). In this retrieval, since there are no model parameters in the state vector,
the model parameter error is zero. The variance due to model parameters is represented in the sensitivity
analysis and becomes zero when we choose a particular set of model parameters. In future
implementations, the model parameters could be included in the state vector and optimized within the
retrieval.

Because the model parameter error goes to zero in our implementation, this means that the
current total retrieval error is the square root of the sum of the smoothing error and the retrieval noise.
The smoothing error is 59.3% to 71.0% of the total retrieval error on average for $CO_2$ and 40.5% to
81.0% of the total retrieval error on average for CO depending on the site and is directly related to the
scaling of the prior covariance matrix. For example, if we did not scale our prior covariance matrix our
smoothing error would be nearly zero. While scaling the a priori covariance matrix increases the
smoothing error, it also results in a reduction to the total retrieval error. The retrieval noise is 29.0% to
40.7% of the total retrieval error on average for $CO_2$ and 19.0% to 59.5% of the total retrieval error on
average for CO depending on the site and has the opposite relationship to the scaling of the a priori
covariance matrix. The retrieval noise reflects the effect of the model covariance matrix that is
composed of the P1 measurement errors and therefore reducing the P1 errors would also reduce the
retrieval noise.

The total retrieval error for each site is determined by the multiplying the retrieved errors by the
site and partial column respective VEM values. After implementing the VEMs, the errors for the lower
partial column $CO_2$ retrieval range from 1.31 ppm to 4.66 ppm and from 0.614 ppb to 1.31 ppb for CO
across all sites and data. As the Caltech and East Trout Lake sites have no validation comparisons, we
use the largest validation error multiplier (that of the lower column Armstrong and upper column Park



Falls comparison) as a higher bound. The comparisons in Fig. 4 are recreated with the scaled errors in Fig. S4.

The retrieval errors are small enough that the TARDISS results have some power for evaluating $CO_2$ fluxes at TCCON sites. The error compared to the overall lower partial column DMF is small, 785  0.80% on average across the five sites for $CO_2$.

| Site | Mean Lower/Upper Column Retrieval Error (ppm for $CO_2$; ppb for CO) | Retrieval Noise (% of total) | Smoothing Error (% of total) | Lower/Upper Column Validation Error Multiplier | Mean Total Lower/Upper Column Error (ppm for $CO_2$; ppb for CO) |
|---|---|---|---|---|---|
| CO₂ Retrievals | | | | | |
| Park Falls | 1.068/0.737 | 29.0 | 71.0 | 1.23/11.7 | 1.31/8.62 |
| Armstrong | 0.977/0.608 | 40.7 | 59.3 | 4.48/8.33 | 4.12/5.06 |
| Lamont | 1.035/0.678 | 32.1 | 67.9 | 3.43/6.29 | 3.08/4.26 |
| Caltech | 1.040/0.670 | 35.0 | 65.0 | 4.48/11.7 | 4.66/7.84 |
| East Trout Lake | 1.025/0.605 | 36.2 | 63.8 | 4.48/11.7 | 4.59/7.08 |
| CO Retrievals | | | | | |
| Lamont | 0.832/0.492 | 59.5 | 40.5 | 1.57/12.5 | 1.31/6.15 |
| Caltech | 0.413/0.105 | 19.0 | 81.0 | 1.57/12.5 | 0.648/1.31 |
| East Trout Lake | 0.391/0.182 | 27.1 | 72.9 | 1.57/12.5 | 0.614/2.28 |

**Table 5.** Errors in the CO and $CO_2$ lower partial column retrievals of each site shown as the average of the entire data time series and broken down into total retrieval error, retrieval noise, smoothing error, 790  validation error multiplier, and total error. The values for total retrieval error and total error are one sigma.

### 3.2.2 Information Content

The information content of the retrieval is determined by the DoF and Shannon information content (H) of the retrieval, each calculated from the averaging kernel of the retrieval. The DoF 795  represent the independent pieces of information that can be retrieved from a measurement. We report our DoF values both normalized by the number of measurements made in a day, as well as the daily average DoF. Since the DoF are calculated as the trace of the averaging kernel, we isolate and report the





DoF from the upper and lower column separately along with the total. The Shannon information content
is a single value to represent the effectiveness of the retrieval to recover information from the model
with respect to the variance in the data. Higher Shannon information content values correspond to a
retrieval with a higher possible effectiveness.

The information content is summarized for each site in Table 6. The overall average lower
column DoF per measurement across all sites and collected data is 0.353 for $CO_2$ and 0.218 for CO. The
lowest DoF average value of 0.287 is in Park Falls and the highest DoF average value of 0.425 is in
Armstrong for $CO_2$ and, between the three sites with CO retrievals, East Trout Lake has the highest
average DoF of 0.292 compared to 0.214 for Lamont and 0.197 for Caltech. The upper column DoF are
significantly less than the lower column DoF due to the constraints implemented in the prior covariance
matrix.

Ideally, DoF values greater than one are desired for traditional profile retrievals. However, the
temporal aspect of our retrieval complicates the discussion. If we consider the $CO_2$ retrievals, the five
sites used in this work made an average of 173 measurements per day so that the DoF value average of
0.353 per measurement still retrieve an average of 60.4 independent pieces of information about the
lower partial column per day which provides significant information on the diurnal variation and the
fluxes into and out of the lower column.

The information content shown in the DoF are mirrored in the Shannon information content.
Similar to the DoF, Park Falls has the lowest and Armstrong has the highest Shannon information
content on average for $CO_2$. These differences are likely driven by the combination of the P1 retrieval
errors and how well the chosen prior covariance matrix matches the temporal aspects of local
meteorology, such as cloud cover or upper tropospheric transport, or the magnitude and time scales of
the local carbon fluxes in the boreal forest and the Mojave Desert. For CO, the East Trout Lake retrieval
has the highest DoF and Shannon information content of the three sites. Interestingly, the Lamont
retrieval has a higher DoF but the Caltech retrieval has a higher average Shannon information content.
While the differences in Shannon information content and DoF between sites are not necessarily
directly comparable, these differences also might be due to the P1 retrieval errors and how well the
chosen prior covariance matrix constrains the solution. Since the Shannon information content includes
the off diagonal terms of the averaging kernel matrix, the larger information content at the Caltech site
suggests that the chosen model covariance matrix and prior covariance matrix are an effective constraint
on the Caltech retrieval.






| Site | Total Degrees of Freedom per measurement (per day) | Lower Column DoF per measurement (per day) | Upper Column DoF per measurement (per day) | Average Measurements per day | Shannon Information Content per day |
|---|---|---|---|---|---|
| CO$_2$ Retrievals | | | | | |
| Park Falls | 0.370 (43.1) | 0.287 (35.9) | 0.0830 (7.19) | 116 | 28.7 |
| Armstrong | 0.495 (112) | 0.425 (98.9) | 0.0700 (13.3) | 227 | 81.0 |
| Lamont | 0.415 (64.9) | 0.337 (55.5) | 0.0782 (9.35) | 159 | 44.4 |
| Caltech | 0.440 (77.8) | 0.364 (67.4) | 0.0760 (10.5) | 180 | 54.0 |
| East Trout Lake | 0.447 (81.6) | 0.354 (70.2) | 0.0925 (11.4) | 182 | 34.8 |
| Overall | 0.433 (75.9) | 0.353 (60.4) | 0.0799 (10.3) | 173 | 48.6 |
| CO Retrievals | | | | | |
| Lamont | 0.247 (30.5) | 0.214 (27.0) | 0.0333 (3.52) | 119 | 18.5 |
| Caltech | 0.197 (39.0) | 0.188 (37.5) | 0.0091 (1.57) | 189 | 23.5 |
| East Trout Lake | 0.292 (55.5) | 0.253 (49.9) | 0.0333 (5.59) | 178 | 25.0 |
| Overall | 0.246 (41.7) | 0.218 (38.2) | 0.0273 (3.56) | 162 | 22.3 |

**Table 6.** Degrees of freedom per measurement (and per day) for the lower column, upper column, and total retrieval, in addition to the Shannon information content separated by site for the CO and CO$_2$ retrievals.




### 3.2.3 Long-term Comparisons

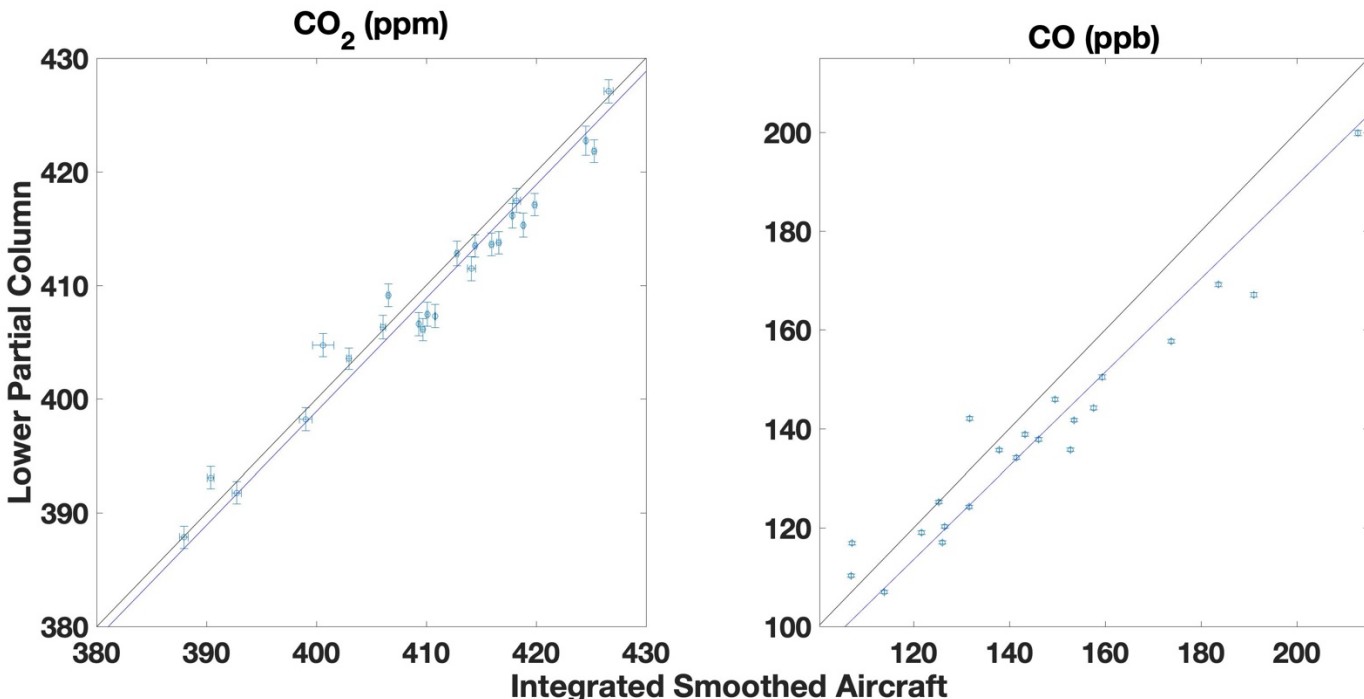

**Figure 6.** East Trout Lake site direct comparisons between the partial column DMF values retrieved from the TARDISS fit and the integrated, smoothed aircraft partial columns for lower column CO$_2$ and CO. The error bars in the x-direction are the reported errors from the aircraft data smoothed the same way as the in situ measurements and the error bars in the y-direction are the output errors from the TARDISS fit. The black solid line is the 1-1 line and the blue line is the linear fit of the data with the y-intercept forced through zero.

In addition to the aircraft and AirCore validation data that include profile measurements at altitudes in the upper troposphere and lower stratosphere, we compare to aircraft data obtained at the Lamont and East Trout Lake sites. We use data obtained between the surface and 7 km from 26 of the 40 flights for CO2 and 26 of the 39 flights for CO made between 2017 and 2020 at East Trout Lake. We also use data obtained between the surface and 6 km from 282 of the 399 flights performed at the Lamont site over the time period of 2008 to 2018 and all 34 flights for CO made between 2017 and 2021. Measurements at both sites were made as part of the NOAA GGGRN aircraft program. Figure 6 (East Trout Lake) and Figure 7 (Lamont) show the retrieved lower partial column DMF plotted against the integrated, smoothed, in situ columns similar to Fig. 4. These measurements were made more frequently but do not include enough high-altitude measurements to compare with our retrieved upper





partial column values, so we use them as an independent comparison to our validation data for our
     lower column $CO_2$ and CO retrievals.

          The consistency of the statistical parameters using the larger number of measurements in the
     long-term comparisons further reinforces the use of the validation comparison results across the entire
     retrieval dataset. Using the operational retrieval parameters, the long-term comparisons have similar
informational and error statistics to the validation comparisons. For the Lamont site, the $CO_2$ retrievals
     have an average DoF per measurement of 0.334 and the comparison slope is 0.995. The overall VEM
     and mean total error calculated from the long-term comparisons is 3.15 and 3.20 ppm compared to 3.43
     and 3.60 ppm from the Lamont, lower column validation comparisons. The Lamont CO retrievals have
     an average DoF per measurement of 0.215 and the comparison slope is 1.0003. The overall VEM and
mean total error calculated from the long-term comparisons is 1.64 and 0.595 ppb compared to 1.57 and
     1.31 ppb from the Lamont, lower column validation comparisons. For the East Trout Lake site, the $CO_2$
     retrievals have an average DoF per measurement is 0.372 and the comparison slope is 0.997. The long-
     term VEM and mean total error for $CO_2$ are 2.42 and 2.48 ppm compared to the Armstrong VEM of
     4.48 used for East Trout Lake and the resulting total error of 4.59 ppm. For long-term CO retrievals at
East Trout Lake, the DoF per measurement is 0.263 and the comparison slope is 0.946. The overall
     VEM and mean total error calculated from the long-term comparisons is 22.5 and 8.34 ppb compared to
     the Lamont VEM of 1.57 used for East Trout Lake and the resulting total error of 0.614 ppb. Some of
     the in situ profile comparisons occur during times with larger CO DMFs that suggest influences from
     sources not accounted for by the P1 a prior profiles such as wild fires which likely resulted in the large
VEM for the long-term CO comparisons.





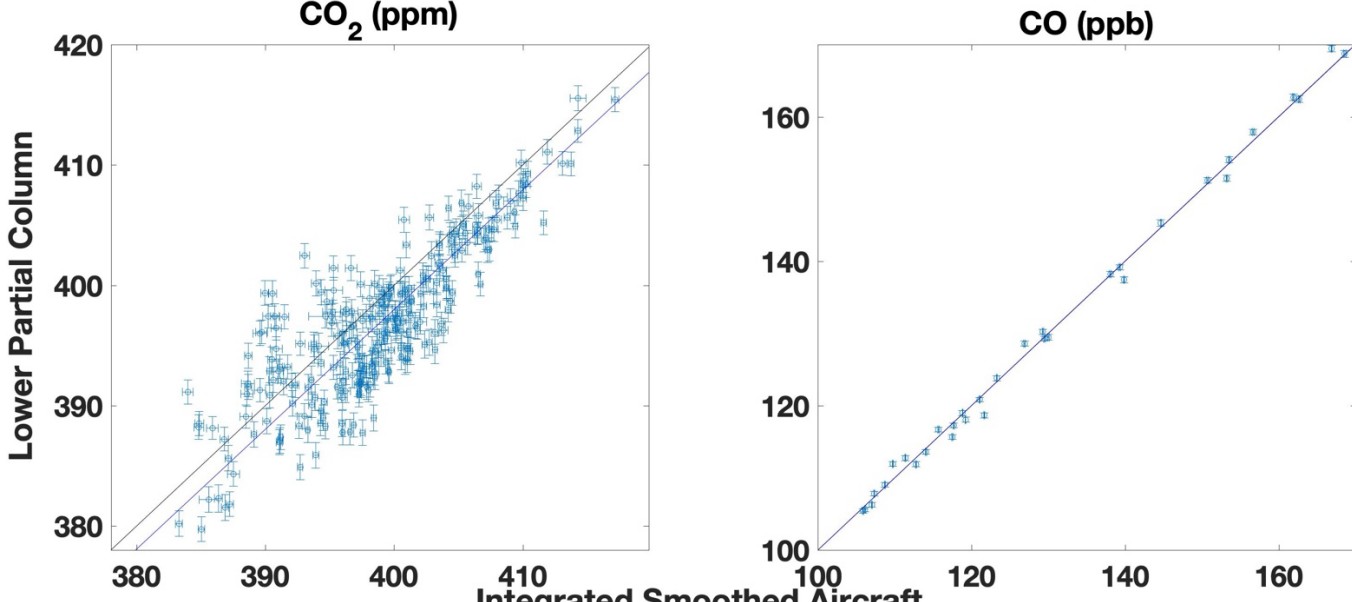

**Figure 7.** Lamont site direct comparisons between the partial column DMF values retrieved
from the TARDISS fit and the integrated, smoothed aircraft partial columns for lower column $CO_2$ and
CO. The error bars in the x-direction are the reported errors from the aircraft data smoothed the same
way as the in situ measurements and the error bars in the y-direction are the output errors from the
TARDISS fit. The black solid line is the 1-1 line and the blue line is the linear fit of the data with the y-
intercept forced through zero.

The informational content of the retrieval helps us understand the algorithm more thoroughly but
could also serve as a diagnostic parameter to indicate the effectiveness of the retrieval for a particular
day of measurement. Figure 8 shows the long-term comparisons between the retrieved lower partial
column and the smoothed, integrated, in situ data at the Lamont site color-coded by the DoF per
measurement for each point. The comparisons with higher DoF per measurement generally sit closer to
the 1-to-1 line as we would expect and suggest that days with higher DoF per measurement would have
a lower associated VEM. Figure S5 shows the VEM calculated when filtering out measurement days
that have DoF per measurement values below a specific threshold. The lowest VEM is calculated when
filtering out days with DoF per measurement lower than 0.35 which excludes roughly half of the 282
flights used. A threshold higher than 0.35 reduces the number of measurements enough that it is no
longer representative of the dataset. As a first step, the data could be filtered for low DoF or low
Shannon information content. Moving forward, the information content could be used to create more
dynamic VEM values for our datasets and provide more precise error values than the conservative,
static VEM per site reported in Table 5. In addition to the DoF, the Shannon information content can be
used to filter retrieval days where the







**Figure 8.** The same comparison shown in Fig. 7 is shown here without error bars and color coded by the DoF per measurement for the comparison day retrieval. The blue line below the black 1-to -1 line is the linear fit of the data with the y-intercept forced through zero.


### 3.3 Current Products

The TARDISS algorithm is applicable to any spectra reported as TCCON data with the correct detector requirements (InGaAs for $CO_2$ and both InGaAs and InSb for CO). Overall there are at least nine years of $CO_2$ data at each site in this work and approximately five years of CO data at the East

Trout Lake, Lamont, and Caltech sites.



Figure 9 shows the monthly mean lower and upper partial column data retrieved from spectra taken over the last decade for the five sites discussed in this work. These data show the global seasonal patterns in $CO_2$ in all sites with values in the Armstrong and Lamont traces being the most similar and consistent. The lower column Park Falls and East Trout Lake traces show the local influences on $CO_2$ in

the sharp dips in the summers when the surrounding forest is most photosynthetically active. The lower column Caltech trace shows a consistent urban enhancement over the global trends of ~5 ppm. All five upper column traces are generally consistent with one another and have a ~6 ppm seasonal fluctuation.

Figure 10 shows the monthly median retrieved lower and upper partial column CO data from the East Trout Lake, Lamont, and Caltech site. We observe seasonality at both sites with maximums in the

winter months and minimums in the summer months. The CO lower partial column data from the Caltech site tends to be larger than those from the Lamont site due to the urban enhancement despite the recent decreasing trend but this is muted when using the monthly medians shown here. The East Trout Lake site show influences from the incomplete combustion of wildfires in both the upper partial column CO traces in both 2017 and 2021.


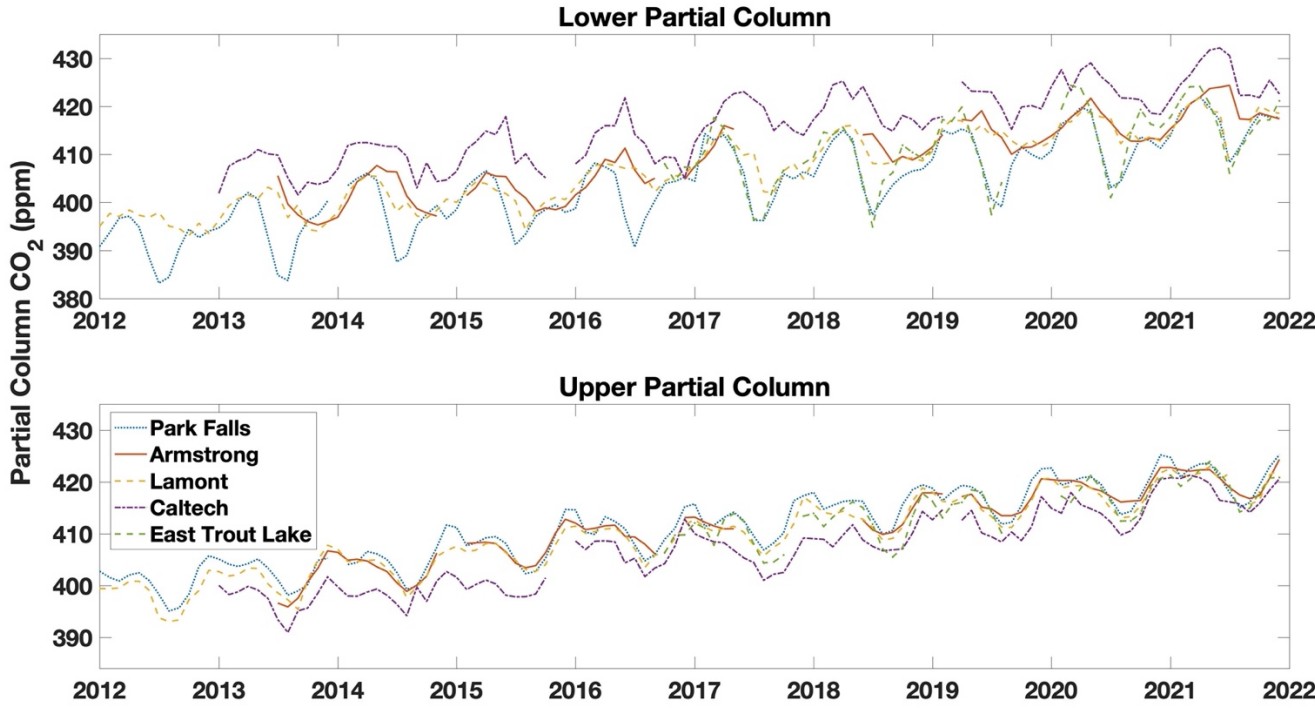



**Figure 9.** Time series plot of the monthly median lower (top) and upper (bottom) partial column values
of $CO_2$ in ppm for the five sites used in the work from 2012 (or the start of measurement) to the end of
2021. Data from before 2012 measured in Park Falls are not used due to instrument alignment issues.

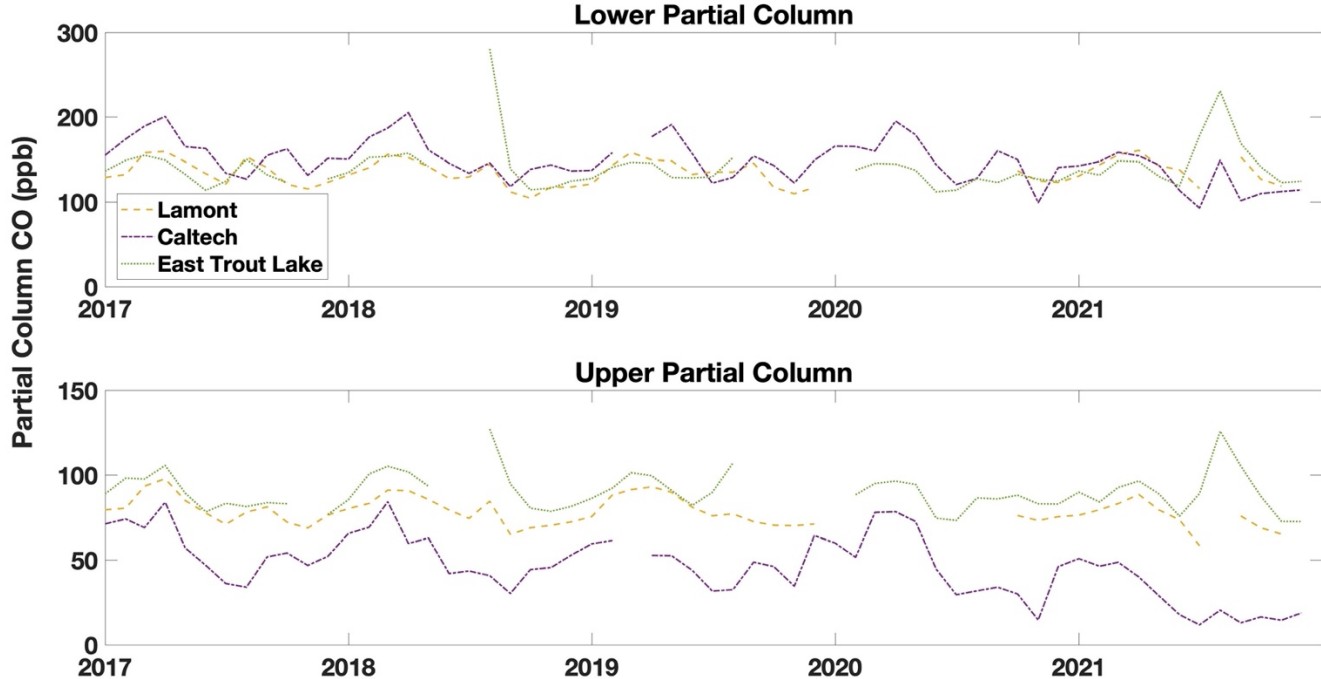


**Figure 10.** Time series plot of the monthly median lower (top) and upper (bottom) partial column
values of CO in ppb for the three sites used in the work that have the InSb detector from 2017 to the end
of 2021. CO has been declining in most of the US cities due to emissions control technologies.

### 3.4 Future Applications

Using the lower partial column data product, we can analyse carbon fluxes in novel ways. The
data retrieved from the Park Falls site is collocated with in situ $CO_2$ data from a ~400-meter tall tower
that measures continuously (Andrews et al., 2014) and also reports eddy covariance fluxes (Berger et
al., 2001). Our lower partial column values are comparable to the in situ data in magnitude and general
trend, particularly during the midday when turbulent mixing is often strong enough to create a more
homogenous mixed layer $CO_2$ concentration (Xu et al., 2019).  In future work, we will compare the
tower $CO_2$ concentration and flux values with estimates of the diurnal variation in partial column $CO_2$.



The lower partial column CO are useful for comparison with other column averaged pollution tracers such as aerosol optical depth (AOD) and TROPOMI $NO_2$. For example, the chemical composition of the atmosphere in the South Coast Air Basin (SoCAB) continues to rapidly change
(Parrish et al., 2016; Van Rooy et al., 2021) and it is of interest to diagnose whether the relationships between primary VOC emission sources, meteorology, and aerosols are also evolving. In future work, we will examine these relationships over the entire partial column data record using both the FTS data and observations from a nearby AERONET AOD measurement. Preliminary comparisons of afternoon average lower column CO values versus afternoon average 500 nm AOD values for days in which the
afternoon average temperatures were above 25 degrees Celsius for the 2016 to 2021 time period show a strong correlation, particularly when decoupled from temperature and accounting for atmospheric water.

**Appendix A: Daily Anomaly Calculation**

Daily anomalies in this work mean the difference between the column values at a particular solar
zenith angle in the afternoon and the column value at the same solar zenith angle in the morning. This approach removes air mass dependencies and allows for a direct comparison of the measured change in column values over a particular day. Due to the differences in the averaging kernel of each window, spectral windows sensitive to different parts of the atmosphere return different total column Xgas values and the ratio of the daily anomalies measured with the different windows used provides insight into how
to weight the different inputs in the inversion. Since the sensitivities of each spectral window is determined by spectroscopy, the daily anomaly ratios are expected to be the same and independent of measurement location.



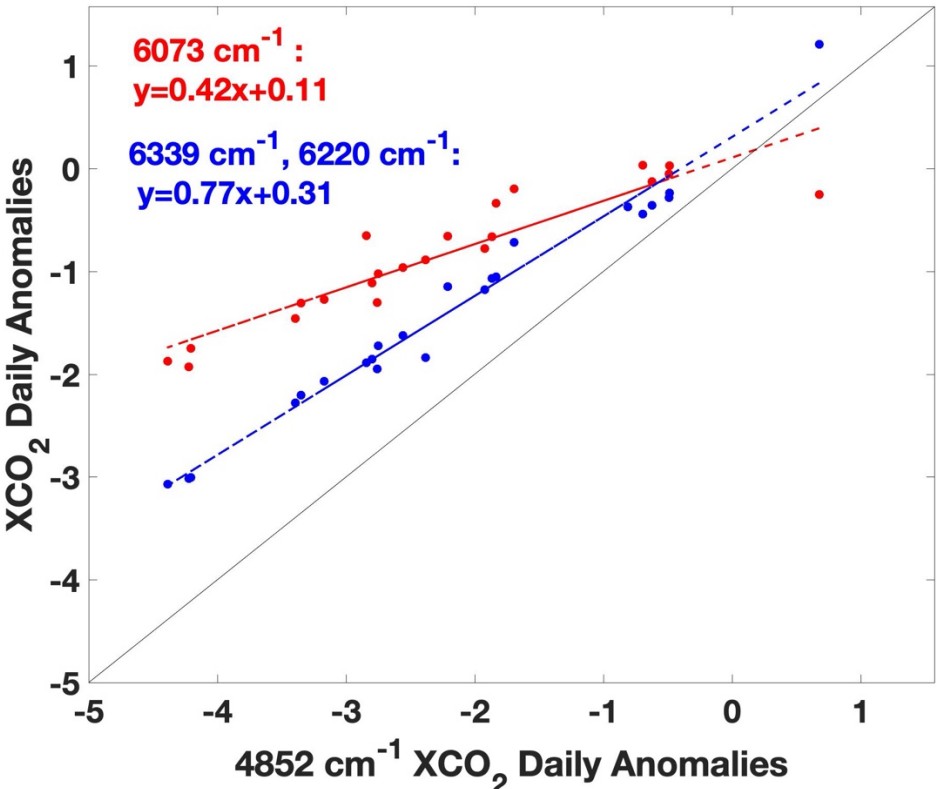

**Figure A.** Scatter plot of the daily anomaly values for the standard 6220 $cm^{-1}$ and 6339 $cm^{-1}$ TCCON windows and the 6073 $cm^{-1}$ $CO_2$ window plotted against the daily anomaly values from the 4852 $cm^{-1}$ CO2 window for days in July, 2018. The black line is the 1-to-1 line and the least squares linear fits for the respective windows are shown in the text on the plot. The slopes of the fit are used as weightings in the TARDISS retrieval.

## Appendix B: Temporal Assimilation

We want to test the influence of the number of observations included in a single retrieval. To do this, we compare the retrieved error value for one measurement fit on its own and with an increasing number of observations until we retrieve with a full day. We take the midday observation from the Park Falls site on July 18, 2018 and retrieve the partial column error values using the least squares method and the maximum a posteriori method (using a static ideal prior scalar to avoid influences from the least squares approach). These values are represented by the points that correspond with zero on the x axis of Fig. B for both the lower and upper partial column errors. We then retrieve the errors of the midday measurement again including the observation before and after it which is represented by the points that correspond with 2 on the x axis of Fig. B. We repeat this method, expanding the number of observations included until we are fitting the entire day of observations.





The left-hand plot of Fig. B1 shows the decrease of the retrieved upper and lower partial column error of the midday point as the number of observations included in the retrieval increases. The upper partial column errors decrease more than the lower partial column errors partially due to the temporal constraints of the a priori covariance matrix. On the contrary, the right-hand plot of Fig. B shows that the inclusion of more observations in the least squares fit does not change the retrieved partial column errors of the midday measurement. Moreover, the partial column errors retrieved using the least squares method are at least six times larger than the partial column errors retrieved using the MAP method. This is due to the use the a priori covariance matrix in the MAP method that can improve upon the best estimate retrieval of the least squares method.

To understand the influence of the prior covariance matrix (overall scaling and temporal constraints), we compare the error values of the least squares method with the MAP method with an entirely uninformed prior covariance matrix. Shown in Fig. B2, the uninformed MAP approach returns errors of similar magnitude to the least squares method. This suggests that a main value of the MAP approach is the use of external information to improve and inform the retrieval.

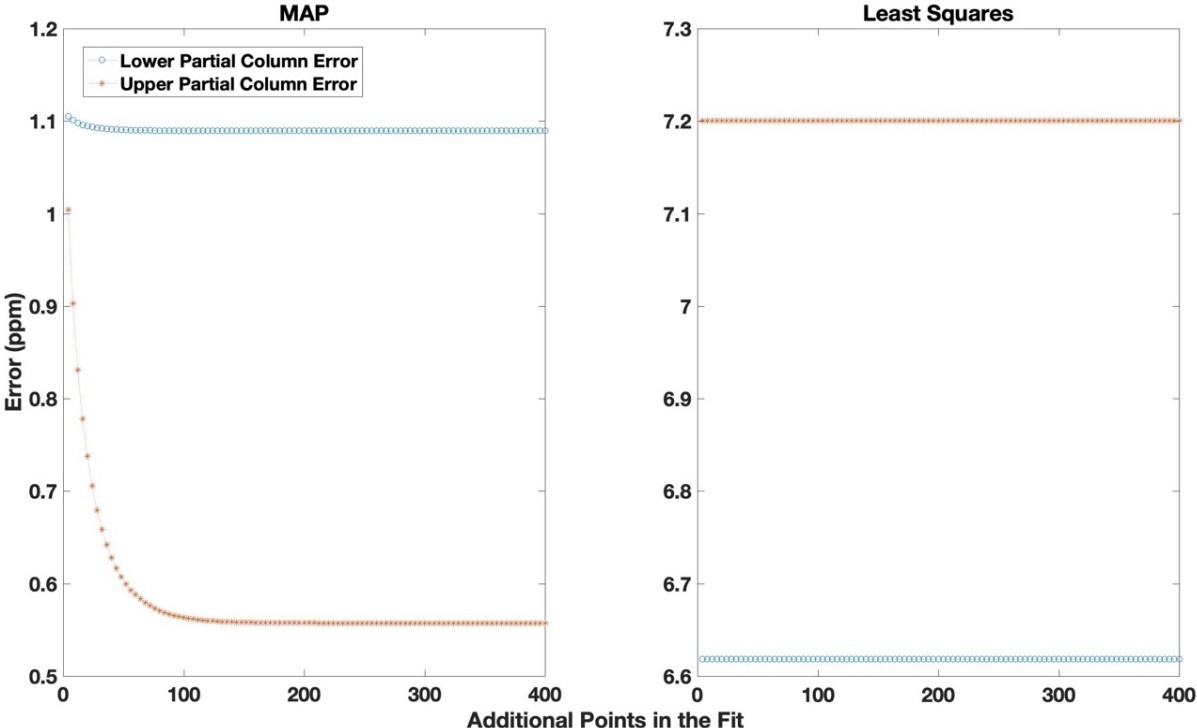

**Figure B1.** Errors in the retrieval of $CO_2$ from the midday total column measurement at the Park Falls site on July 18, 2018 using the MAP method outlined by equation 12 and the least squares method outlined by equation 18. The blue circles represent the error in the lower partial column and the orange asterisks represent the error in the upper partial column. Note the difference in the range of the y axis in





the left and right plots both of which are in parts per million. The x axis indicates the number of points included in the overall fit with zero additional points representing the retrieval of a single spectra.

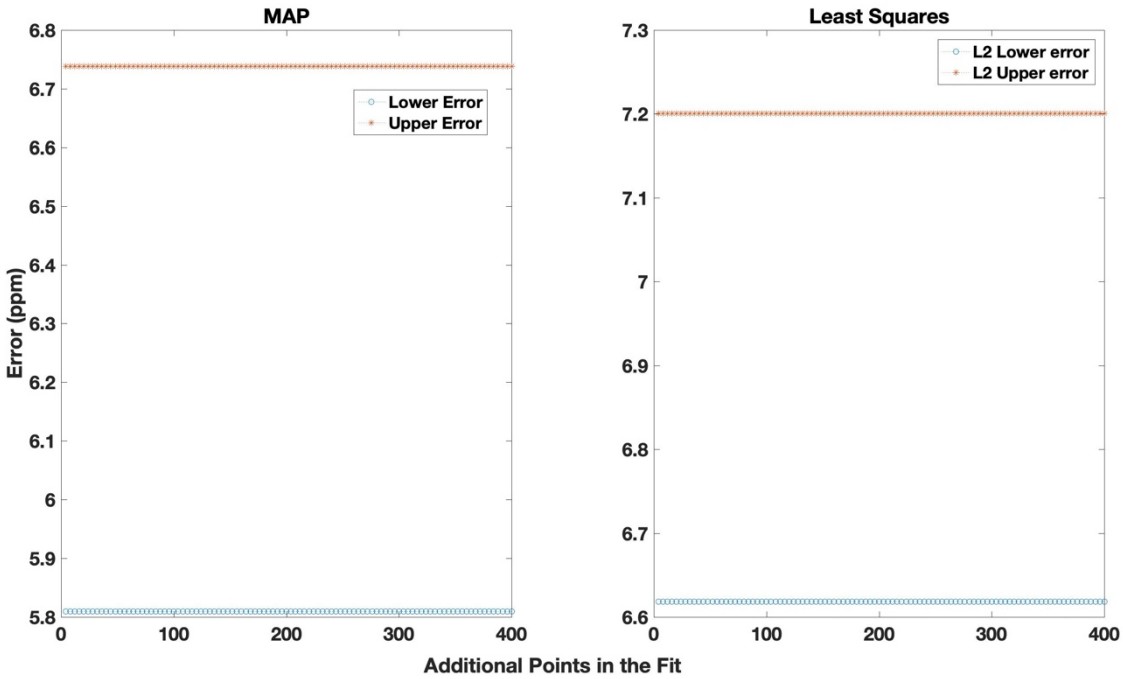

**Figure B2.** Same as Fig. B1, except the prior covariance is removed from the MAP retrieval.

**Acknowledgements**

We thank NASA via 80NSSC22K1066 for support of retrievals from the TCCON stations. A portion of this research was carried out at the Jet Propulsion Laboratory, California Institute of Technology, under a contract with the National Aeronautics and Space Administration (80NM0018D0004). The authors would like to thank the ObsPack scientists for the use of the in situ profiles used for validation that were gathered during the various campaigns. The data are downloaded from https://gml.noaa.gov/ccgg/obspack/data.php?id=obspack_co2_1_GLOBALVIEWplus_v5.0_2019-08-12 and were most recently accessed on September 2nd, 2022. In particular, we would like to thank the NASA LaRC AVOCET and DACOM groups for the KORUS-AQ $CO_2$ and CO data, respectively; NASA Goddard for the Picarro $CO_2$ data at the Armstrong AFB; the SEAC4RS and ATom groups for the $CO_2$ data; the NOAA Global Monitoring Division for the AirCore $CO_2$ and CO data, and the long term aircraft $CO_2$ and CO at the SGP ARM site and the ETL site. We thank Jochen Stutz for their effort in establishing and maintaining Caltech AERONET site.


## Data Availability

The data used in this study are made up of TARDISS retrieval products from five TCCON stations. The retrieval data are publicly available through CaltechDATA (https://doi.org/10.22002/pn9de-cry27) and the data input into the retrieval are publicly available via https://tccondata.org/. Retrieval code is currently available by request.

## Author Contributions

HP wrote the TARDISS algorithm following an approach suggested by PW. HP retrieved the data with it and prepared the paper with thorough feedback from the coauthors. JL developed the theoretical framework for the TARDISS algorithm. CR retrieved the TCCON data using GGG for the Lamont, Caltech, and Park Falls sites. GCT gave input on the retrieval algorithm. DW gave input on the validation data and method. LTI and JRP maintain the Armstrong site. KM and BB provided insight and in situ data for the validation. All authors contributed to the review and editing of the work.

## Competing Interests

The authors declare they have no conflicts of interest.

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
