# Peer review of "Inferring the vertical distribution of CO and CO2 from TCCON total column values using the TARDISS algorithm"

_Atmospheric Measurement Techniques, 2022_

## Author Response (AR1)

**Authors' Response to Reviews of**

**Inferring the vertical distribution of CO and CO₂ from TCCON total column values using the TARDISS algorithm**

Harrison A. Parker, Joshua L. Laughner, Geoffrey C. Toon, Debra Wunch, Coleen M. Roehl, Laura T. Iraci, James R. Podolske, Kathryn McKain, Bianca C. Baier, Paul O. Wennberg

We appreciate the work of the referees and their helpful comments and would like to thank the referees for their help in improving this manuscript. Responses to the comments are below.

As a note, between the first submission and now, a bug was discovered that reduced the value of $z_{a,TCCON}$ from Equation 1 (now Equation 3) in the algorithm code so the retrievals and comparisons were rerun and the corresponding plots, tables, and values within the manuscript were corrected. Correcting the bug also removed the need for Appendix A in the original submission since applying additional weighting to the windows used in the retrieval degraded the performance of the fit. We have also moved the Appendix B from the original submission to the SI as S1 in the current version of the text.

- Anonymous reviewer #1:

General Comments:

Reviewer: I had major issues understanding in detail the concepts proposed by the paper. This refers in particular to how all the equations fit together and what quantities the variables represent. Without being in the inner circle of TCCON discussions, the methodology (section 2.2, equations scattered throughout the paper and appendix, equations and concepts taken from other papers, TCCON jargon) is hard to follow. I recommend that the authors make a serious effort to present the methodology in a more concise, clearer, yet complete way, that is more accessible to the general reader. Some information might be missing (formulae for calculating smoothing and noise errors). Some of the very detailed discussions in the results section could be summarized.

Authors: We moved all the equations into section 2.2 and segmented section 2.2 into five subsections to improve the readability of the paper. Each subsection has been edited to be more informative and more transparent about the motivation for each equation and their relation to the overall algorithm. The equations used from other

papers were included explicitly and explained further to improve the accessibility. Changes to the retrieval code removed the need for the previous Appendix A which should also concentrate the methodology discussions. We have included the formulas for the smoothing error and retrieval noise in the Error Calculations section. We have also included a flowchart for the processes starting at the measured spectrum and ending with the TARDISS output data and characterization to help with the accessibility of the concepts and the transition from the single measurement, single spectral window calculations to the full day, multiple spectral window matrices. We have also included Table S1 with a full list of the variables used in the work, their descriptions, and the equation they were defined by if they were defined by an equation.

Reviewer: What is the key difference with respect to previous work by Roche et al., 2021 that makes performance better here? The two approaches are in the end quite similar, both generating vertical information by combining windows with different sensitivities. Whether the combination is realized during spectral analysis or a posteriori should not matter in principle. Is the claimed better performance here because the columns for each window are scaled individually to the same WMO standards such that line strength inconsistencies are corrected?

Authors: The main difference between the TARDISS approach and a more traditional spectral profile retrieval is that spectral measurements and a priori meteorological profiles that would be considered subpar for direct spectral fits for profile information could still be used in the TARDISS algorithm to infer partial column information. By using total column DMF values from spectral fits that are corrected in post-processing procedures to align with WMO standards, the TARDISS approach removes the issues of fitting with inconsistencies between different spectral windows. Further, restricting the fit to the differences between total column DMF values seems to eliminate the issues of oscillation or unphysical DMF deviations. This also limits the informational content to be retrieved but the temporal aspect of the algorithm allows for partial column information to be inferred from multiple measurements. Our analysis shows that the use of external, a priori, temporal information is also helpful in constraining the retrieval and improving validation performance. We elaborate on these ideas in Section 1, 2.2, 3.2, and 3.4.1 and have further discussion in Section 4.

Reviewer:  I do not understand equation (18) and the related discussion.

Authors: We moved Equation 18 (now Equation 27) to section 2.2.5 and added more information about the equation and its motivation. The validation error multipliers calculated by Equation 27 are applied to the output errors of the TARDISS retrieval on a

site-by-site basis so that the reported error values reflect the performance of the retrieval in the validation comparisons. This approach gives a more conservative value for the retrieval errors since the retrieval is effectively limited by the amount it can scale the partial columns. If a calculated validation error multiplier is less than one, the retrieved error is the conservative error value and a validation error multiplier of one is used instead.

Reviewer: Equation (12) calculates the MAP solution. What is the least-squares solver of equation (18) used for? Does it refer to the usage of the least-squares solution as the prior for the MAP solution? If this is the case, it implies that the inversion works reasonably well in an (unconstrained) least-squares sense. Why would one then want to go through all the MAP machine (which lowers the degrees of freedom for signal)? It generally appears incompatible with the idea of MAP that a least-squares solution is taken as prior.

Authors: The least-squares approach does return a reasonable solution; however, the associated errors are too large to be used for any sort of scientific purposes as they are on the order of 10 ppm for $CO_2$. The use of the scaled a priori covariance matrix drastically reduces the retrieval errors through the scaling and through the temporal constraint. The MAP approach does have reduced degrees of freedom of signal but, even with the largest constraints on the fit, there are at least enough degrees of freedom for the retrieval of a lower partial column value per hour for $CO_2$ and ~4 per hour for CO. The future use of curated a priori information in the a priori covariance matrix, a priori partial column scalars, or additional fitting parameters could improve the retrievals and their retrieved degrees of freedom.

Reviewer: The validation data mostly come from airborne or tower in-situ measurements which do not cover the entire vertical column. How is the missing part of the column (either at the bottom or at the top) taken into account when comparing to the (lower or upper) partial column from the TCCON measurements? What are related uncertainties? I would think that a careful consideration is important since 1) the CO2 (maybe also CO) vertical profile is most variable in the lowest few hundred meters (i.e. extrapolating from or into the lowest few hundred meters is error prone) and 2) the targeted accuracy for CO2 partial columns is on the ppm level.

Authors: We have adjusted the accounting for the parts of the profile that are not measured by in situ methods to include errors based on the variability designated by the parts of the profile that are measured and added further description of this in the

text in Section 3.1. This is the approach taken for TCCON comparisons for the total column comparisons and is applied to the partial column comparisons. The long-term comparisons with low altitude in situ measurements have errors that are much larger than previously reported although the errors in the smoothed, in situ partial column $CO_2$ values are still much smaller than the reported errors in the retrieval.

Reviewer: From the validation study (e.g. Fig. 4, table 4, Fig. 6, 7), I find it hard to evaluate whether the performance of the proposed algorithm is convincing or not. It would need, for reference, comparisons to the performance of the TCCON standard retrievals i.e. taking the P1 scaling factors and calculating upper and lower partial columns and comparing those to the validation dataset as well. Probably, showing performance of the priors would also be interesting.

Authors: We have included the validation comparisons of the P1 (now called TCCON for simplicity) individual spectral window partial columns in Fig. 5 and Table 4 and a discussion of the results in Section 3.3.1. For the most part, the TARDISS retrieved partial column values have an improved direct validation comparison and with improved precision. The lower partial column CO and upper partial column $CO_2$ retrievals both improve the comparison slopes and reduce the mean ratio deviation compared to the individual window comparisons. The lower partial column $CO_2$ retrievals greatly improve the direct comparison but are slightly less precise which is reflected in the increased retrieval errors. The upper partial column CO retrievals have comparisons that are within the comparisons of the individual windows and are not improved. These comparisons suggest that the TARDISS retrieval is performing as intended and is improving the accuracy and precision of most partial column values.

Specific Comments:

Reviewer: L30: Is Doppler broadening really the limiting factor in the troposphere – as opposed to pressure broadening or temperature modulating the population of rotational levels?

Authors: We have removed the implication that Doppler broadening is a limiting factor of profile retrieval in the troposphere.

**Reviewer:** L86f: The discussion of MOPITT appears misplaced. The vertical information for the satellite instrument MOPITT comes from combining thermal emission and absorption (Schwarzschild equation) while, for direct-sun measurements (Beer-Lambert's law) such as TCCON, vertical information comes much more indirect through line shapes and relative optical depths. If MOPITT is discussed, these conceptual differences should be highlighted. Plus, there is similar work on GOSAT and a range of other TIR satellites (IASI, AIRS, ...) e.g. Kulawik et al., 2017, https://doi.org/10.5194/acp-17-5407-2017; Kuze et al., 2022, https://doi.org/10.1016/j.rse.2022.112966.

**Authors:** We have removed the flawed discussion of MOPITT from the introduction to focus on the ground-based CO profile retrievals performed by NDACC.

**Reviewer:** Equ. 4: $x_{a,i}$ needs subscript P1, I guess?

**Authors:** We have changed the P1 notation to TCCON for simplicity and we have added the TCCON subscript to the necessary terms in Equation 4 (now Equation 7).

**Reviewer:** Equ. 17: Please explain what "Avert" is and motivate the equation (what is the "star" operator?).

**Authors:** We have added further description and reasoning for the Avert calculation and removed the oversight of the asterisk as an operator.

**Reviewer:** L308: The Xgas notation is undefined, I think.

**Authors:** We have removed the Xgas terms to reduce jargon where possible and defined it where it is used.

**Authors' Response to Reviews of**

**Inferring the vertical distribution of CO and CO₂ from TCCON total column values using the TARDISS algorithm**

Harrison A. Parker, Joshua L. Laughner, Geoffrey C. Toon, Debra Wunch, Coleen M. Roehl, Laura T. Iraci, James R. Podolske, Kathryn McKain, Bianca C. Baier, Paul O. Wennberg

We appreciate the work of the referees and their helpful comments and would like to thank the referees for their help in improving this manuscript. Responses to the comments are below.

As a note, between the first submission and now, a bug was discovered that reduced the value of $z_{a,TCCON}$ from Equation 1 (now Equation 3) in the algorithm code so the retrievals and comparisons were rerun and the corresponding plots, tables, and values within the manuscript were corrected. Correcting the bug also removed the need for Appendix A in the original submission since applying additional weighting to the windows used in the retrieval degraded the performance of the fit. We have also moved the Appendix B from the original submission to the SI as S1 in the current version of the text.

Anonymous reviewer #2:

General Comments:

Reviewer: Reading the subject of this paper, the first thing I wanted to see was the comparison between validation data and TCCON partial column results (P1) vs the (presumably improved) comparison to TARDISS, but no such comparison is ever made. I have no basis to judge how much (or indeed if) the algorithm succeeds.

Authors: We have included the validation comparisons of the TCCON individual spectral window partial columns in Fig. 5 and Table 4 and a discussion of the results in Section

3.3.1. For the most part, the TARDISS retrieved partial column values have an improved direct validation comparison and with improved precision. The lower partial column CO and upper partial column $CO_2$ retrievals both improve the comparison slopes and reduce the mean ratio deviation compared to the individual window comparisons. The lower partial column $CO_2$ retrievals greatly improve the direct comparison but are slightly less precise which is reflected in the increased retrieval errors. The upper partial column CO retrievals have comparisons that are within the comparisons of the individual windows and are not improved. These comparisons suggest that the TARDISS retrieval is performing as intended and is improving the accuracy and/or precision of most partial column values.

**Reviewer:** Section 2.2 is the heart of the formulation. It is more difficult to read than it needs to be. Many of the symbols need more precise explanation, and sometimes have inconsistent descriptions. A few examples (not an exhaustive list) follow. In eq. (2), $\mathbf{x}_{a,P1}$ is called a 'profile' while $\mathbf{x}_{part}$ is a 'partial column'; a study of eq. (3) is needed to understand the symbols in (2) (they are both profiles). In another case, eq. (17) is presented without justification and the actual meaning of $\mathbf{A}_{vert}$ is opaque.

**Authors:** We have divided section 2.2 into five subsections, moved all the equations into section 2.2, and included further description and motivation for the equations to improve readability and accessibility of the concepts. We have fixed the oversight of referring to $\mathbf{x}_{part}$ as a partial column and now refer to it as a profile for the use in Equation 2 (now Equation 5). We added more description of the transition between Equation 2 and 3 (now Equation 5 and 6) to allow for each equation to be accessible independently. We also include further description of the Avert term and its motivation to clarify its importance in converting the temporal sensitivities of the retrieval into vertical sensitivities that can be used to compare smoothed in situ profiles to our partial column data. We have also included a flowchart for the processes starting at the measured spectrum and ending with the TARDISS output data and characterization to help with the accessibility of the concepts and the transition from the single measurement, single spectral window calculations to the full day, multiple spectral window matrices. We have also included Table S1 with a full list of the variables used in the work, their descriptions, and the equation they were defined by if they were defined by an equation.

Reviewer: In eq.(1), $z_{a,P1}$ is said to be the median value of the TCCON-retrieved scale factor in the set of windows used, times the original a priori column (L. 226). Twelve pages later (L. 501-2) it turns out to be the *daily* median for $CO_2$, and equal to 1 for CO, which modified my understanding of the intervening material, and required re-reading. Also in eq.(1) $\mathbf{x}_{part}$ is called 'partial column' while $\mathbf{x}_{a,P1}$ is called the 'profile.'

Authors: We have changed the terminology in lines 501-2 (now lines 531-2) to "The value of the a priori scalar for the lower and upper partial column scalar ($x_{a,\gamma}$ in Equation 16) is the least squares solution for the respective column ($x_{L2}$ in Equation 15)." to clarify that the a priori term discussed is the a priori partial column scalar choice and that $z_{a,TCCON}$ is the median value of the TCCON scale factors and is independent of a priori partial column scalar choice. We have also fixed the error of referring to $\mathbf{x}_{part}$ as a partial column and now refer to it as a profile with added description to improve understandability.

Reviewer: More generally, the authors have clearly made extensive analyses of the algorithm for various choices of input and data from various sites, which is commendable, and report these results in exhaustive detail in tables and figures. Unfortunately, descriptions in the text of the results shown in the tables and figures are more detailed and extensive than is useful, and inhibit identification and understanding of the key results.

Authors: We have removed much of the extensive reporting of the results from the texts and included further discussion in Section 3. We have also included a conclusions section to summarize the key results of the manuscript.

Specific Comments:

Reviewer: L.174 'United States' includes a site in Saskatchewan!

Authors: We have corrected "United States" to "North America."

Reviewer: Fig. 3 caption: 'The profile above 6 km not shown' does not seem to refer to anything.

Authors: We corrected the caption to the correct value of 10 km.

Reviewer: L.527 contains 'of the' twice

Authors: We removed the duplicated words.

Reviewer: P.36 ends in the middle of a sentence which is not continued on the next page.

Authors: We removed the unfinished sentence.

Reviewer: Connor et al., 2008 is cited (it's the first citation in the text) but it's not in the reference list

Authors: We added the Connor et al., 2008 to the reference list.

Reviewer: L.990 'Fig. B'; which one?

Authors: We clarified the reference to be to Fig. B1 (now Fig. S1).

---

## Author Response (AR2)

**Authors' Response to Reviews of**

**Inferring the vertical distribution of CO and CO₂ from TCCON total column values using the TARDISS algorithm**

Harrison A. Parker, Joshua L. Laughner, Geoffrey C. Toon, Debra Wunch, Coleen M. Roehl, Laura T. Iraci, James R. Podolske, Kathryn McKain, Bianca C. Baier, Paul O. Wennberg

We appreciate the work of the referees and their helpful comments and would like to, again, thank the referees for their help in improving this manuscript. Responses to the comments are below.

- Anonymous reviewer #1:

Specific Comments:

Reviewer: L81: "They report errors near 2%" ... which is actually not so much worse than the errors reported here implying that the retrievals from the MIR might be useful for flux inversions after all.

Authors: We have changed the phrase "to limit their use for carbon cycle studies" to "to encourage the exploration of other methods for use for carbon cycle studies" to focus on the motivation for TARDISS.

Reviewer: L231-L234: Is this obsolete? Equation (2) defines all this.

Authors: L231 – L 234 has elements defined elsewhere, however we feel it is necessary to be explicit with the steps that deal with grouping so that there is less ambiguity about the main terms in the TARDISS algorithm.

Reviewer:  L272: K boldface

Authors: We have bolded the character.

Reviewer: L286: y boldface

Authors: We have bolded the character.

Reviewer: L294, equ (15): Shouldn't it be "x_L2-x_a" on lhs, or "x_a+…" on the rhs?

Authors: We have changed the equation.

Reviewer: L312: "forward mapping matrix" previously introduced as "Jacobian matrix"

Authors: We changed the terminology to be consistent.

Reviewer: L366: G boldface

Authors: We have bolded the character.

Reviewer: L395, L400, L404: The "errors" are probably the "square roots" of the variances in the matrices, not the variances per se.

Authors: We have corrected the description of the errors as they are the square roots of the variances.

Reviewer: L713: the -> they

Authors: We have corrected the term.

Reviewer: Fig.8: Why not using pressure as vertical axis as in Fig. 1?

Authors: We have changed the vertical axis to match Fig. 1.

**Reviewer:** L870: "While scaling the a priori matrix by a higher value increases the smoothing error …" Isn't it the other why around: loose prior constraint implies less smoothing error.

**Authors:** We have clarified the statement to "using a more constrained a priori covariance matrix increases the smoothing error".